# Dysfunctional adipocytes promote tumor progression through YAP/TAZ-dependent cancer-associated adipocyte transformation

Yaechan Song [1,4], Heeju Na[1,4], Seung Eon Lee[1], You Min Kim[1], Jihyun Moon[1], Tae Wook Nam[1], Yul Ji[2], Young Jin[1], Jae Hyung Park[1], Seok Chan Cho[1], Jaehoon Lee [1,3], Daehee Hwang[2], Sang-Jun Ha [1], Hyun Woo Park [1], Jae Bum Kim [2] & Han-Woong Lee [1,3] ✉

Obesity has emerged as a prominent risk factor for the development of malignant tumors. However, the existing literature on the role of adipocytes in the tumor microenvironment (TME) to elucidate the correlation between obesity and cancer remains insufficient. Here, we aim to investigate the formation of cancer-associated adipocytes (CAAs) and their contribution to tumor growth using mouse models harboring dysfunctional adipocytes. Specifically, we employ adipocyte-specific BECN1 KO (BaKO) mice, which exhibit lipodystrophy due to dysfunctional adipocytes. Our results reveal the activation of YAP/TAZ signaling in both CAAs and BECN1-deficient adipocytes, inducing adipocyte dedifferentiation and formation of a malignant TME. The additional deletion of YAP/TAZ from BaKO mice significantly restores the lipodystrophy and inflammatory phenotypes, leading to tumor regression. Furthermore, mice fed a high-fat diet (HFD) exhibit decreased BECN1 and increased YAP/TAZ expression in their adipose tissues. Treatment with the YAP/TAZ inhibitor, verteporfin, suppresses tumor progression in BaKO and HFD-fed mice, highlighting its efficacy against mice with metabolic dysregulation. Overall, our findings provide insights into the key mediators of CAA and their significance in developing a TME, thereby suggesting a viable approach targeting adipocyte homeostasis to suppress cancer growth.

The study of tumor progression extends beyond cancer cells alone and encompasses the complex ecosystem surrounding them. This ecosystem, referred to as the tumor microenvironment (TME), comprises various cell types, including immune, epithelial, and endothelial cells, fibroblasts, and adipocytes. Adipocytes, in particular, constitute a considerable portion of stroma in breast, colorectal, and endometrial tumors[1,2]. Although immune cells and fibroblasts have been extensively investigated, research on adipocytes as primary components of a TME is only emerging. Adipocytes at the tumor protruding region

actively exchange signals with cancer cells, leading to dynamic morphological and functional transformations[2]. These adipocyte populations, known as cancer-associated adipocytes (CAAs), exhibit dedifferentiated phenotypes and fibroblast-like characteristics[3,4] and undergo metabolic adaptation in a nutrient-restrictive environment[5]. CAAs also secrete adipokines and signaling molecules to provide a pro-tumorigenic environment[6]. Moreover, CAAs are prominently activated to release lipid metabolites, such as free fatty acids (FFAs), through lipolysis. Cancer cells utilize these FFAs as a source of fuel through

[1]Department of Biochemistry, College of Life Science and Biotechnology, Yonsei University, Seoul 03722, Republic of Korea. [2]Department of Biological Sciences, Seoul National University, Seoul 08826, Republic of Korea. [3]Gemcro, Inc, Seoul 03722, Republic of Korea. [4]These authors contributed equally: Yaechan Song, Heeju Na. ✉e-mail: hwl@yonsei.ac.kr

fatty acid oxidation (FAO), facilitating their uncontrolled growth[7,8]. Given that certain types of cancer are highly populated with adipocytes, it has become increasingly important to characterize CAAs in order to understand the development of malignant TME.

The adipose tissue of an obese individual is subjected to hypoxic conditions due to the increased number and size of adipocytes. These adipocytes secrete pro-inflammatory factors, such as TNFα, interleukin-6 (IL-6), monocyte chemoattracted protein-1 (MCP1), and CXC motif chemokine ligand 12 (CXCL12), leading to local and systemic inflammation[9]. Dysfunction of these adipocytes can disrupt endocrine signaling, thereby exacerbating diverse illnesses, such as diabetes, cardiovascular diseases, hepatic steatosis, and atherosclerosis[10]. However, while several studies have demonstrated a close relationship between obesity and poor prognosis in patients with cancer[11–13], the specific mechanisms by which dysfunctional adipocytes contribute to tumor progression remain unresolved.

Autophagy is a cellular process responsible for the degradation and recycling of cellular components, which is crucial for maintaining cellular homeostasis. Beclin1 (BECN1) is an autophagy-related gene that plays a vital role in autophagy initiation by activating the class 3 phosphoinositide 3-kinase (PI3K) complex[14]. The relationship between BECN1 and cancer progression has garnered significant interest due to its involvement in membrane nucleation and response to cellular stress. Clinical data suggests that BECN1 has tumor-suppressive functions, as monoallelic BECN1 deletion has been found in approximately 30% of breast cancer and 77% of patients with ovarian cancer[15–17]. While its role as an autophagy inducer can explain some of the intrinsic functions of BECN1 in tumor progression, recent studies have also highlighted various autophagy-independent functions of BECN1. For instance, Hu et al. reported that BECN1 could lead to the suppression of colorectal cancer metastasis in mice through inhibition of STAT3 phosphorylation via Janus kinase 2 (JAK2) in an autophagy-independent manner[18].

The Hippo signaling pathway is a crucial process for maintaining cellular homeostasis and modulating the proliferation and differentiation of cells. Yes-associated protein (YAP) and its paralog transcriptional coactivator PDZ-binding motif (TAZ) act as cofactors for TEAD, a major transcription factor in the Hippo pathway. YAP/TAZ are phosphorylated by LATS1, which is activated by MOB1/2 and MST1/2. Upon phosphorylation, the translocation of YAP/TAZ into the nucleus is hindered; instead, they are sequestered in the cytoplasm, predisposed to undergo proteasomal degradation. A recent study has reported that YAP/TAZ activity in adipocytes is crucial for determining their differentiation state[19]. These studies provide evidence that the Hippo pathway plays a role in mediating adipocyte dedifferentiation to generate CAAs.

Previously, we generated adipocyte-specific Becn1 KO (BaKO) mice to investigate the role of BECN1 in mature adipocytes. Our findings revealed that depletion of BECN1 in adipocytes result in the development of dysfunctional phenotypes, enhancing inflammatory signals and accumulation of ER stress. Hence, BaKO mice exhibit impaired adipose tissue function, leading to severe lipodystrophy phenotypes[20]. Herein, we aim to investigate the impact of dysfunctional adipocytes on tumor progression. Implementing our mouse models and BECN1-deficient adipocyte cell lines, we identify the key factors that facilitate the enhanced tumor progression within adipose-rich environments. Our results show that the loss of BECN1 result in adipocyte transformation, displaying characteristics similar to CAAs. Transformation of these adipocytes involve activation of YAP/TAZ signaling and inflammatory response. Additionally, we demonstrate that MOB1, as a key regulator of the Hippo pathway, plays a role during the differentiation and dedifferentiation of adipocytes. These findings highlight the significance of BECN1-mediated regulation of YAP/TAZ in the adipocyte-TME and suggest that inhibition of YAP/TAZ to maintain adipocyte homeostasis could provide a viable therapeutic approach to target a malignant TME.

## Results

### BaKO mice harbor dysfunctional adipocytes, which promote tumor progression in breast and colon cancer models

To explore the impact of dysfunctional adipocytes in the TME, we employed BaKO mice to assess the progression of five syngeneic tumor models from various tissue origin. We found that colorectal (MC-38) and breast cancer (EO771) cells showed accelerated growth in BaKO mice (Fig. 1a, b; Supplementary Fig. 1a, b), while the melanoma and lung cancer cells did not (Supplementary Fig. 1c–e). Next, we utilized the mammary tumor virus polyoma middle T antigen (MMTV-PyMT) transgenic mouse model that spontaneously develops breast cancer[21]. Here, we generated adipocyte-specific Becn1 KO conditions during breast cancer development by crossing MMTV-PyMT mice with BaKO mice (PyBaKO). PyBaKO mice exhibited an increased tumor incidence and progression (Fig. 1c, d), resulting in higher lung metastasis and mortality rates (Fig. 1e, f).

We previously reported that BaKO mice developed lipodystrophy, liver steatosis, and glucose intolerance[20]. These metabolic dysregulations are known to promote tumor progression through systemic impacts[22,23], which could have contributed to the rapid tumor growth observed in BaKO mice. Therefore, to determine the direct effects of BECN1-deficient adipocytes on cancer growth, we established immortalized stromal vascular cell (imSVC) lines with conditional Becn1 KO system (ROSA-Cre^ERT2). imSVCs were fully differentiated into adipocytes and then treated with 4-hydroxytamoxifen (4-OHT) to deplete BECN1. Next, we co-injected WT or Becn1 KO imSVCs and breast cancer cells (4T1) into the mouse flanks (Fig. 1g, h). Consistently, tumors grew more rapidly as the number of Becn1 KO adipocytes in the mix increased, which was not observed in cancer cells mixed with WT adipocytes (Fig. 1i). These results demonstrate that Becn1 KO adipocytes are sufficient to promote tumor growth without the systemic effects observed in BaKO mice.

### BECN1-deficient adipocytes induce pro-inflammatory signals such as TNFα and LCN2, to support tumor growth

To elucidate the functional consequences of BECN1 depletion in adipocytes, we conducted RNA-sequencing analysis (RNA-seq) on BaKO inguinal WAT (iWAT). Primarily, we performed hallmark gene set enrichment analysis (HGSEA) to identify the altered gene sets in BaKO WAT. The adipose tissue in BaKO was accompanied by enrichment of multiple gene expression associated with TNFα signaling and inflammatory response (Fig. 2a; Supplementary Fig. 2a). Further examination revealed activation of TNFα downstream signaling in Becn1 KO adipocytes (Fig. 2b). Additionally, BECN1-deficient adipocytes secreted elevated levels of TNFα into the cultured media, comparable to those seen in normal adipocytes treated with lipopolysaccharide (LPS) (Fig. 2c).

TNFα, has been shown to impair adipocyte differentiation, adipogenic potential, and fat storage[24]. Notably, adipocytes co-cultured with cancer cells displayed a marked increase in TNFα signaling (Supplementary Fig. 2b), which can regulate the expression of various adipokines. Therefore, we tested whether secretory factors produced by BECN1-deficient adipocytes are sufficient to promote cancer cell growth. Treatment with concentrated conditioned media (CM) extracted from BECN1-deficient adipocytes led to accelerated growth of both MC-38 and EO771 compared to that with CM derived from WT adipocytes (Fig. 2d, e).

Adipokines are signaling molecules secreted by adipocytes that play critical roles in regulating cellular processes, such as cell proliferation, angiogenesis, and inflammation, all of which are essential in tumor growth and metastasis[2,25,26]. To further investigate adipokine

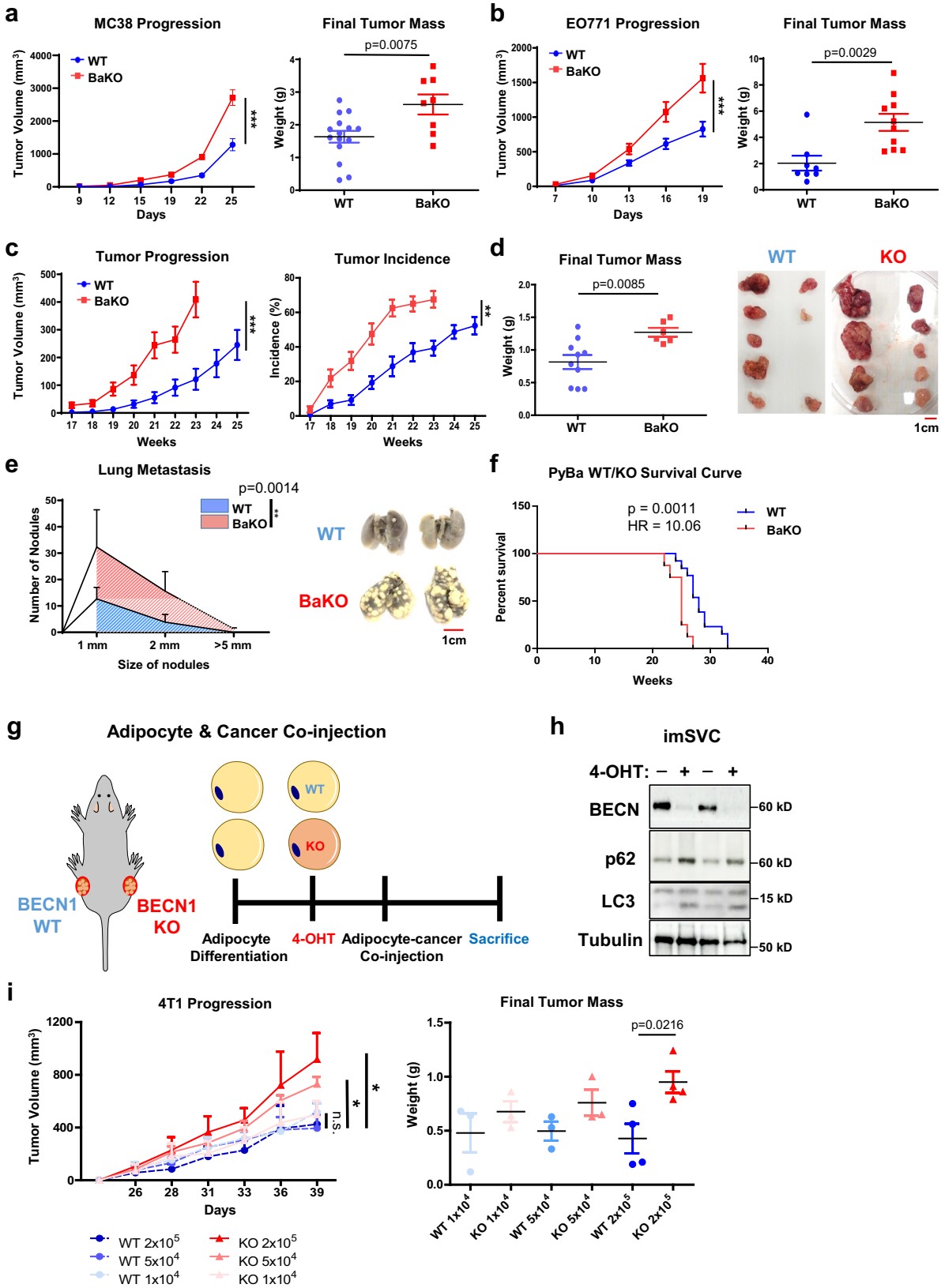

secretion and its potential role in tumor growth, we measured the levels of 38 adipokines in BaKO plasma using a mouse adipokine array kit (Fig. 2f). We then quantified the expression of each adipokine and aligned them in the order of significance and found that BaKO plasma contained higher levels of oncogenic cytokines, such as SerpinE1, Resistin, Lipocalin2 (LCN2), and insulin-like growth factor binding proteins (IGFBPs) (Fig. 2f). Since these adipokines could be secreted from multiple organs systemically, we inspected their expression in WT and BECN1-deficient adipocytes. Among multiple adipokines, LCN2 was consistently upregulated in BECN1-deficient adipocytes (Fig. 2g, h). Subsequently, we confirmed that adipocyte-LCN2 could be amplified with recombinant TNFα[27], suggesting that constitutively

**Fig. 1 | Dysfunctional adipocytes promote tumor progression in breast and colon cancer models. a** Tumor volumes and weights after subcutaneous injection of MC-38 into 6- week- old WT and adipocyte specific BECN1 KO (BaKO) mice (WT, n = 15; BaKO, n = 8). **b** Tumor volumes (WT, n = 18; BaKO, n = 12) and weights (WT, n = 8; BaKO, n = 10) after mammary fat pad injection of EO771 into 6-week-old WT and BaKO mice. **c** Average growth kinetics and incidence of breast tumors generated from 10 mammary glands of MMTV-PyMT BECN1 WT mice (PyWT, n = 16) and MMTV-PyMT BECN1 KO mice (PyBaKO, n = 16). **d** Average tumor mass isolated from each mammary gland and a representative image of the tumors from PyWT (n = 10) and PyBaKO (n = 6) mice. Mice were sacrificed at 23 weeks of age. **e** Metastasized tumor nodules counted from the lungs of PyWT (n = 13) and PyBaKO (n = 8) mice at 23 weeks of age. Representative image was taken from mice sacrificed at 23 weeks

of age. **f** Kaplan–Meier plot of survival curve of PyWT and PyBaKO mice. Log-rank test $p$-value ($p$) and hazard ratio (HR) between PyWT (n = 13) and PyBaKO (n = 8) mice are indicated. **g** Schematic timeline of adipocyte-cancer co-injection. (4-OHT: 4-hydroxytamoxifen). **h** Western blotting analysis of protein lysates isolated from immortalized stromal vascular cells (imSVCs) before being mixed with cancer cells. **i** Tumor volumes and weights after subcutaneous injection of 4T1 cells mixed with WT or Becn1 KO adipocytes. Mice were sacrificed on the 39th day following co-injection (n = 3,4). Statistics were calculated using two-tailed unpaired Student's $t$-test (**a**, **b**, **d**, **i**), ordinary two-way ANOVA (**a**, **b**, **c**, **e**, **i**), and Kaplan–Meier survival curve analysis (**f**). Data are shown as mean ± SEM; *$p$ ≤ 0.05, **$p$ ≤ 0.01, ***$p$ ≤ 0.001. Western blot results are representatives of at least three independent experiments.

active TNFα signaling in BaKO iWAT could be responsible for LCN2 secretion (Fig. 2i; Supplementary Fig. 2c).

LCN2 has been implicated in the development of metabolic disorders[28], and its levels are upregulated in the adipose tissue of obese individuals with low-grade inflammation[29]. To this end, LCN2 is recognized as a pro-inflammatory signal that promotes cancer cell survival and malignancy[30]. In line with these studies, treatment of recombinant LCN2-protein successfully increased the proliferation of MC-38 and EO771 cells in a dose-dependent manner (Supplementary Fig. 2d, e). Next, we assessed the impact of LCN2 uptake from cancer cells co-cultured with BECN1-deficient adipocytes. EO771 cells co-cultured with BECN1-deficient adipocytes exhibited an elevated level of LCN2 receptor (LCN2R) (Supplementary Fig. 2f). Knock-down of the LCN2R in cancer cells was sufficient to attenuate the effect of *Becn1 KO* adipocytes on cancer growth in-vitro (Fig. 2j). These results highlight that BECN1 depletion induces secretion of inflammatory and oncogenic adipokines, especially TNFα and LCN2, contributing to the establishment of a pro-tumorigenic niche.

### CAAs and BECN1-deficient adipocytes share dedifferentiated phenotypes along with activated YAP/TAZ signaling
CAAs are known to acquire fibroblast-like phenotypes while losing their mature adipocyte properties[2,31–33]. When influenced by adjacent tumor cells, CAAs undergo a dynamic transformation to support cancer cell growth (Fig. 3a). Indeed, co-culturing adipocytes with breast cancer cells resulted in remarkably low expression of adipogenic markers and BECN1 (Fig. 3b, c). Similar to BECN1 deficient adipocytes, co-cultured adipocytes expressed higher levels of *Lcn2* (Supplementary Fig. 3a). Therefore, we performed a colony formation assay to assess the impact of adipocytes on cancer cell growth. Our result demonstrates that the growth rate of cancer cells increases proportionally with the number of co-cultured adipocytes (Fig. 3d).

Next, we sought to characterize the CAAs and BECN1-deficient adipocytes in-vivo. We performed RNA-seq to understand the physiological alterations that WATs undergo upon interaction with cancer cells. We isolated the peritumoral region where adipose tissue surrounds the tumor mass and performed HGSEA with naive WAT (Fig. 3e, f). RNA-seq results from naive, peritumoral, and BaKO WATs further highlighted the resemblance between BECN1-deficient adipocytes and CAAs. HGSEA identified that among the 23 upregulated and 8 downregulated gene sets in peritumoral and BaKO WATs, they shared 10 and 3 gene sets to be up- and down-regulated when compared to naive WAT, respectively (Fig. 3g).

Zhu et al. observed that adipocytes located near cancer cells undergo dedifferentiation under restrictive metabolic conditions, contributing to the generation of a tumor-supportive microenvironment through adipocyte mesenchymal transition (AMT)[34]. Our HGSEA results revealed that epithelial-mesenchymal transition (EMT) was one of the commonly upregulated gene sets in both BaKO and peritumoral WATs (Supplementary Fig. 3b). In concordance, the genes associated with adipogenesis and fatty acid (FA) metabolism were consistently

downregulated, supporting the concept of mesenchymal transition of adipocytes (Fig. 3h, i)[32,34].

In addition to EMT, we found YAP signatures to be also commonly upregulated in both BaKO and peritumoral WATs (Fig. 3j). GSEA on 'C6: oncogenic signature gene sets' revealed that YAP signaling was a prominently upregulated gene set in BaKO (Supplementary Fig. 3c, d). In fact, YAP/TAZ levels were significantly elevated in BaKO iWAT (Supplementary Fig. 3e), and YAP/TAZ activity was enhanced in both co-cultured and BECN1-deficient adipocytes (Supplementary Fig. 3f, g). Further characterization of BECN1-deficient adipocytes revealed suppression of adipogenic potential and loss of lipid contents due to a marked increase in lipolysis (Supplementary Fig. 3h–j). These characteristics of BECN1-deficient adipocytes are considered vital features of CAAs (Fig. 3a–c)[7,8]. Taken together, the acquisition of the shared phenotypes with CAA enables BECN1-deficient adipocytes to form a favorable milieu for tumor growth.

### Autophagy is dispensable for BECN1-mediated YAP/TAZ activation
We previously showed that the loss of adipocyte-BECN1 resulted in autophagy inhibition[20]. Autophagy in adipocytes plays a crucial role not only in maintaining cellular homeostasis but also in facilitating mitochondrial clearance[35]. To determine whether the autophagy-related function of BECN1 drives adipocyte transformation, we tested the effects of autophagy inhibitors, bafilomycin A and hydroxychloroquine (HCQ), on adipocytes. Although autophagy flux was clearly blocked in mature adipocytes, we could not observe any notable changes in YAP/TAZ expression or its downstream signaling (Fig. 4a; Supplementary Fig. 4a–c).

Autophagy-related gene 7 (*ATG7*) is another key molecule involved in autophagy flux, particularly in autophagosome elongation and closure[36]. While *BECN1* deletion in HEK293 induced nuclear translocation of YAP/TAZ, *ATG7* deletion alone could not replicate this effect (Fig. 4b; Supplementary Fig. 4d). Furthermore, we employed adipocyte-specific *Atg7 KO* mice (ATG7aKO) to assess the tumor progression of MC-38 and EO771. Unlike *BECN1*, *Atg7* deletion in mouse adipocytes was insufficient to provide a pro-tumorigenic environment for MC38 and EO771 (Fig. 4c, d; Supplementary Fig. 4e, f). Consistently, co-culture of HCQ-treated adipocytes and cancer cells could not enhance the growth of EO771 (Supplementary Fig. 4g). Moreover, when mice were treated with HCQ or co-injected with HCQ-treated adipocytes, there was no significant difference in EO771 progression compared to the control groups (Fig. 4e, f). These studies highlight the unique roles of adipocyte-BECN1 in regulating cellular homeostasis beyond autophagy. Collectively, the alteration of YAP/TAZ expression and adipocyte transformation observed in BaKO should be attributed to the autophagy-independent function of BECN1.

### Dynamic YAP/TAZ regulation modulates adipocyte differentiation status
Because adipocyte differentiation involves a dynamic alteration of multiple gene expression, we assessed how the BECN1, YAP1, and TAZ

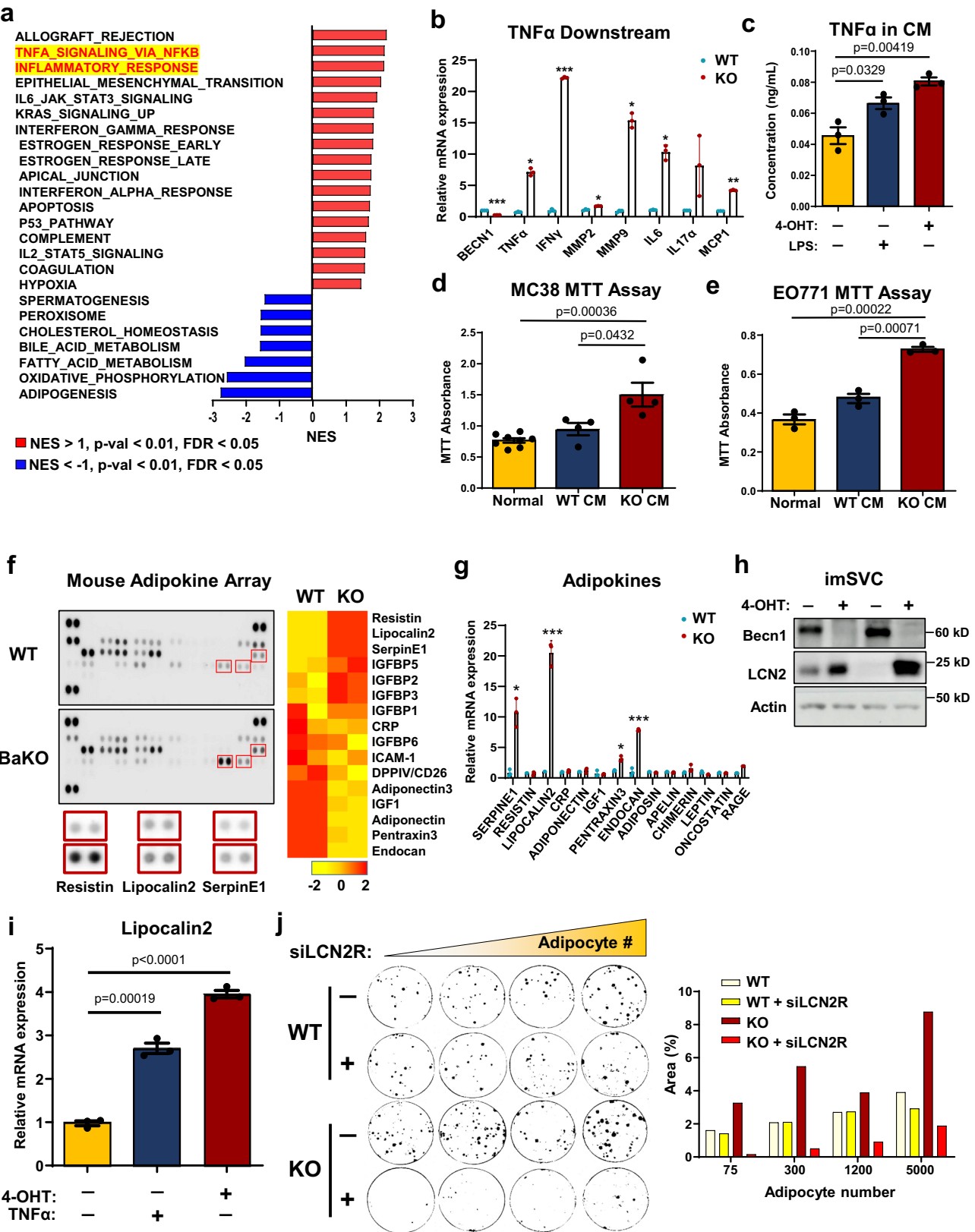

**b** TNFα Downstream

**c** TNFα in CM

**d** MC38 MTT Assay

**e** EO771 MTT Assay

**f** Mouse Adipokine Array

**g** Adipokines

**h** imSVC

**i** Lipocalin2

**j**

levels change throughout the process. We observed that the BECN1 level increases as adipocyte differentiation progresses. In contrast, signaling molecules for cellular plasticity and proliferation, such as YAP/TAZ and β-catenin, were decreased in mature adipocytes (Fig. 5a, b). As we further explored the role of YAP/TAZ in adipocyte maturation, our immunofluorescence (IF) analysis showed that the

adipocytes expressing high PLIN1 levels have low nuclear YAP/TAZ levels, whereas PLIN1-negative preadipocytes contain higher levels of nuclear YAP/TAZ (Supplementary Fig. 5a).

To understand the physiological response of YAP/TAZ under BECN1 depletion, we established doxycycline-inducible *Becn1 KO* adipocytes. We found that YAP and β-catenin levels were inversely

**Fig. 2 | Dysfunctional adipocytes induce pro-inflammatory signals such as TNFα and LCN2, to support tumor growth. a** GSEA plot for RNA-seq result of BaKO iWAT. Significantly altered 'Hallmark gene sets' are aligned according to their normalized enrichment score (NES), p-val, and FDR. **b** Relative mRNA expression of TNFα downstream genes between WT and *Becn1 KO* imSVCs (n = 3, biologically independent samples per group). **c** Concentration of TNFα in the CM extracted from imSVCs of indicated genotypes using ELISA (n = 3, biologically independent samples per group). LPS (100 ng/mL) treatment for 24 h. **d** Cell viability assay performed on MC-38 (Normal n = 8; WT CM n = 4; KO CM n = 4, biologically independent cells per group) and EO771 **e** grown in normal or conditioned media extracted from WT or BECN1 KO adipocytes for 72 h (n = 3, biologically independent cells per group). **f** Mouse adipokine array performed in dot blot and heatmap analysis representing the altered adipokines. Samples were extracted from WT and BaKO mouse plasma (n = 2, each genotype). **g** Relative mRNA expression of adipokines associated with tumor growth in fully differentiated WT or BECN1 KO imSVCs (n = 3, biologically independent samples per group). **h** Western blot analysis of protein lysates isolated from imSVCs. **i** Relative mRNA expression of LCN2 in differentiated imSVCs treated with or without TNFα recombinant proteins for 24 h (n = 3, biologically independent samples per group). **j** Colony assay of EO771 co-cultured with WT or BECN1 KO adipocytes. A total of 50 cancer cells were seeded with or without lipocalin2 receptor siRNA (siLCN2R) treatment for 2 days before the co-culture (co-cultured adipocyte numbers are as indicated). Area (%) was quantified using ImageJ. This experiment was conducted once. Statistics were calculated using two-tailed unpaired Students *t*-test (**b, c, d, e, g, i**). Data are shown as mean ± SEM; *$p \leq 0.05$, **$p \leq 0.01$, ***$p \leq 0.001$. Western blot results are representatives of at least three independent experiments.

correlated with BECN1 levels (Fig. 5c). IF imaging also revealed the enhanced levels of YAP/TAZ and β-catenin in *Becn1 KO* adipocytes (Fig. 5d, e; Supplementary Fig. 5b–d). Consistently, BECN1 depletion in imSVCs enhanced nuclear translocation of YAP/TAZ (Supplementary Fig. 5e). Next, we investigated whether CAAs also expressed altered YAP/TAZ and β-catenin levels. In line with BECN1-deficient adipocytes, cancer cell-co-cultured adipocytes exhibited elevated YAP/TAZ nuclear translocation and β-catenin levels (Supplementary Fig. 5f).

### BECN1 regulates the Hippo pathway through interaction with MOB1

Some studies have introduced the reciprocal interaction between two tightly regulated biological processes: the Hippo pathway and autophagy[37]. To elucidate the underlying mechanism of how BECN1 deficiency leads to the activation of YAP/TAZ signaling, we performed a proximity ligation assay (PLA) and examined the interactions between BECN1 and proteins involved in the Hippo signaling pathway.

Upon adipocyte differentiation, interactions between Hippo-associated proteins, such as MOB1-LATS1 and LATS1-YAP1 were enhanced, indicating the activation of Hippo pathway to control YAP/TAZ levels (Fig. 5f). However, these interactions were significantly diminished by BECN1 depletion (Fig. 5f, Supplementary Fig. 6a). Subsequently, overexpression of BECN1 along with the Hippo-associated proteins, such as YAP1, LATS1, MST1, and MOB1 demonstrated a remarkably strong interaction of BECN1-MOB1 compared to other Hippo-associated proteins (Main Fig. 5g, Supplementary Fig. 6b, c), and the interaction was stronger than that between LATS1 and MOB1 (Supplementary Fig. 6d). To investigate the functional implication of BECN1 on MOB1, we assessed the changes in MOB1 and p-MOB1 levels upon BECN1 overexpression. The results showed increased MOB1 level, enhancing the phosphorylation of YAP (Fig. 5h). Conversely, with BECN1 depletion from adipocytes, MOB1 levels diminished in both imSVCs and BaKO iWAT (Fig. 5i, j). These findings uncover the mechanistic insights into the contribution of BECN1 to the Hippo pathway.

### Lipodystrophy phenotypes in BaKO were restored by additional deletion of YAP/TAZ

To eliminate YAP/TAZ-mediated adipocyte transformation, we crossed BaKO with *Yap/Taz*-floxed mice and generated adipocyte-specific *Becn1/Yap1/Taz* triple-KO mice (BYTaKO) (Fig. 6a; Supplementary Fig. 7a). These mice developed adipocytes with impaired autophagy (Supplementary Fig. 7b); however, lipodystrophy phenotypes observed in BaKO were restored in BYTaKO (Fig. 6b–g). This process involved the restoration of adipose tissue mass, potentially leading to the alleviation of liver steatosis through successful lipid storage in the adipose tissue (Fig. 6b–d; Supplementary Fig. 7c). While lipodystrophy observed in BaKO was prevented in BYTaKO, single *Yap1* or *Taz* deletion was insufficient to restore the BaKO phenotypes (Supplementary

Fig. 7d). Accordingly, transcriptomic analyses revealed that enhanced TNFα signaling observed in BaKO was curtailed in BYTaKO iWAT (Fig. 6e, f). Plasma TNFα concentration was also reduced by the additional deletion of *Yap/Taz* (Fig. 6g), implying alleviated systemic inflammation in BYTaKO. Additionally, BYTaKO mice were resistant to HFD-induced obesity (Supplementary Fig. 7e, f).

Next, we investigated whether additional YAP/TAZ deletion regulates LCN2 in adipocytes. The measurement of LCN2 levels revealed that BYTaKO WAT CM contained less LCN2 than that of BaKO (Fig. 6h). Consistently, LCN2 mRNA expression was diminished in BYTaKO iWAT compared to that in BaKO (Supplementary Fig. 8a). To identify how YAP/TAZ deletion suppresses LCN2 expression, we analyzed the Chromatin Immunoprecipitation Sequencing (CHIP-seq) data of YAP and TEAD available from a public database. The results showed robust peaks at the LCN2 promoter region, while no significant peaks were detected for TNFα (Fig. 6i; Supplementary Fig. 8b). Subsequently, we performed luciferase assay to validate direct regulation of LCN2 and TNFα at the transcriptional level through YAP. Indeed, adding recombinant YAP protein remarkably increased the luciferase activity of LCN2 promoter containing TEAD binding motifs (Fig. 6j, k; Supplementary Fig. 8c, d).

To evaluate whether BECN1/YAP/TAZ (BYT)-deficient adipocytes could impact tumor growth, we generated doxycycline-inducible BYT KO adipocytes and performed a colony formation assay. When grown with BECN1-deficient adipocytes, cancer cells exhibited an enhanced proliferation rate in a dose-dependent manner (Fig. 6l). Conversely, BYT-deficient adipocytes interacted with cancer cells to suppress their growth. Together, these findings suggest that additional *Yap/Taz* deletion prevented adipocytes from providing excessive pro-inflammatory factors and LCN2.

### Inhibition of YAP/TAZ activity suppresses the pro-tumorigenic effect of BECN1-deficient adipocytes and HFD feeding

To validate the effect of BYT-deficient adipocytes in vivo, we transplanted MC-38 and EO771 into WT, BaKO, and BYTaKO. Consistent with transcriptomic analysis and colony formation assay, tumor growth was significantly suppressed in BYTaKO compared to that in BaKO (Fig. 7a, b; Supplementary Fig. 9a, b). We next explored the relevance between each mouse genotype, including the peritumoral WATs. To gain a preliminary understanding of the gene expression patterns in WATs, we performed principal component analysis (PCA) on RNA-seq results of naive and peritumoral WATs (WT, BaKO, and BYTaKO). A significant difference was observed between BaKO and WT naive WATs, whereas the BYTaKO WAT exhibited a milder phenotype. This difference was even more pronounced in peritumoral WATs, suggesting that BYTaKO WAT underwent a milder transformation upon cancer transplantation (Supplementary Fig. 9c). As reported previously, the transformation of adipocytes near cancer cells is associated with the downregulation of mature adipocyte markers[4,32,38] (Fig. 3b, h). Accordingly, the expression of adipogenesis- and FA

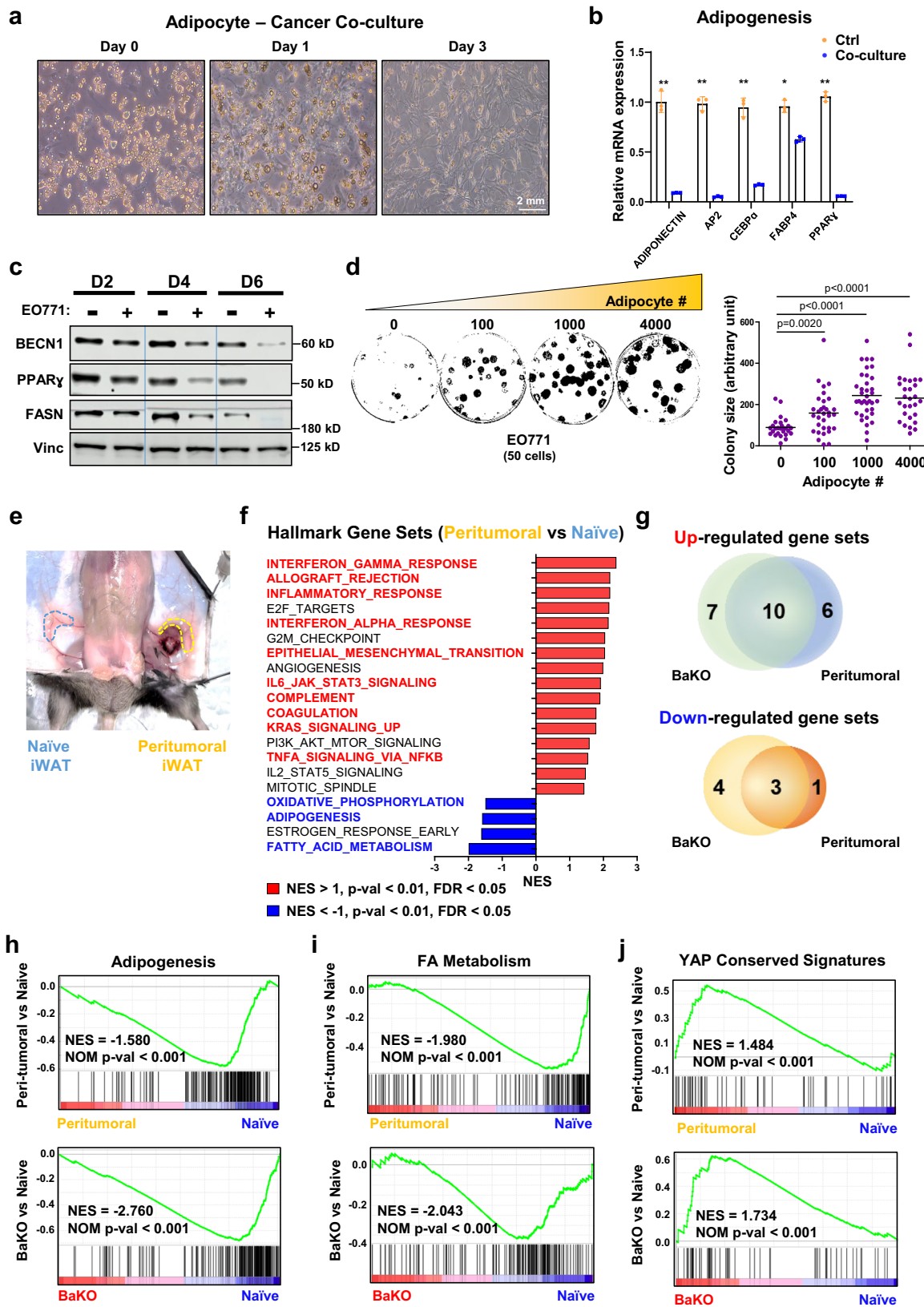

metabolism-associated gene sets indicated that the adipocytes in peritumoral WAT of all genotypes had lost adipogenic potential and mature adipocyte properties (Fig. 7c). We observed that BaKO WATs exhibited significantly lower levels of adipogenesis- and FA metabolism-associated genes, whereas BYTaKO WATs retained relatively higher levels of these genes (Fig. 7c).

Several studies have demonstrated the anti-tumor activity of verteporfin (VP) through suppression of YAP/TAZ signaling[39,40]. To complement our genetic mouse model (BYTaKO), we employed VP to inhibit YAP/TAZ activity in adipose tissue (Supplementary Fig. 9d, e). To avoid the direct effect of VP on cancer cells, we administered a lower dose of VP that does not affect WT tumor

**Fig. 3 | Cancer-associated adipocytes exhibit similar phenotypes to BECN1-deficient adipocytes. a** Differential interference contrast microscopy image of adipocytes co-cultured with EO771 for the indicated duration. **b** Relative mRNA expression of adipogenic genes from differentiated 10T1/2 adipocytes co-cultured with EO771 breast cancer cells for 4 days (n = 3, biologically independent samples per group). **c** Western blotting analysis of protein lysates isolated from differentiated 10T1/2 adipocytes co-cultured with EO771 breast cancer cells over the time course. **d** Colony assay of 50 EO771 cells co-cultured with the indicated number of adipocytes. Each colony size was measured using imageJ (50 distinct colonies per group). This experiment was conducted once. **e** Image of naive and peri-tumoral

iWAT samples resected from a mouse. **f** GSEA result of BaKO and naive adipose tissue ranked by NES. Gene sets that also appear to be differentially expressed in peritumoral WAT are colored in red and blue. **g** Venn diagram of GSEA results on BaKO and peritumoral WATs compared to naive adipose tissue; p < 0.01, FDR < 0.05, NES < −1 or NES > 1 (**h**) GSEA plots for adipogenesis, (**i**) FA metabolism, and (**j**) YAP conserved signaling gene sets compared between naive, peritumoral, and BaKO WAT. Statistics were calculated using two-tailed unpaired students $t$-test (**b**, **d**). Data are shown as mean ± SEM; *$p \leq 0.05$, **$p \leq 0.01$, ***$p \leq 0.001$. Western blot results are representatives of at least three independent experiments.

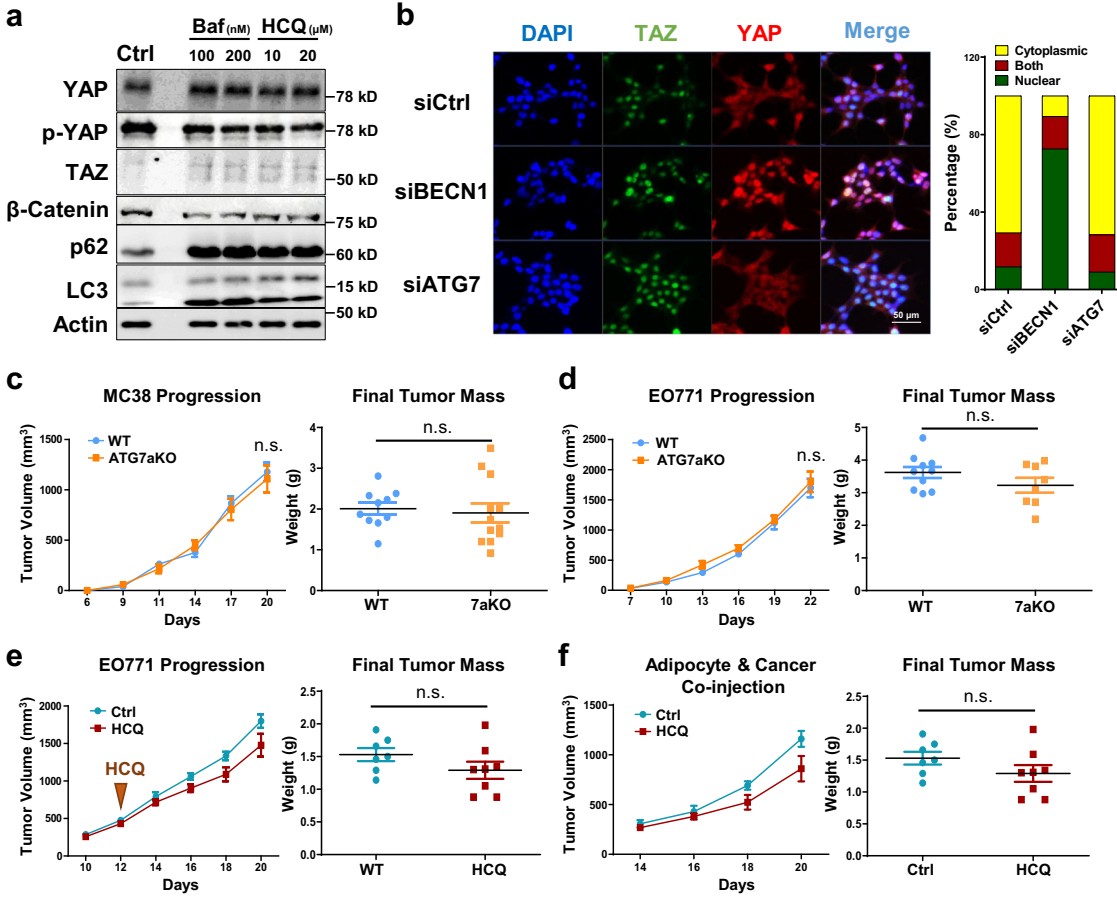

**Fig. 4 | Autophagy inhibition in adipocytes is insufficient to induce YAP/TAZ signaling and tumor growth, while BECN1 regulates the Hippo pathway through interaction with Mob1. a** Western blot analysis of protein lysates isolated from differentiated 3T3L1 treated with autophagy inhibitors, Bafilomycin A (Baf) and Hydroxychloroquine (HCQ), for 48 h. **b** Representative IF image of HEK293T cells stained with YAP, TAZ, and DAPI after siRNA BECN1 and siATG7 transfection. Localization of YAP/TAZ was quantified using ImageJ. Refer to Supplementary Fig. 4d for the Western blot analysis. **c** Tumor volumes and weights measured after subcutaneous injection of MC-38 into 7-week-old WT and ATG7aKO (WT, n = 10; ATG7aKO, n = 12). **d** Tumor volumes and weights measured after

mammary fat pad injection of EO771 into 7-week-old WT and ATG7aKO (WT, n = 10; ATG7aKO, n = 8). **e** Tumor volumes and weights measured after mammary fat pad injection of EO771 into 7-week-old mice treated with and without HCQ (50 mg/kg, injected everyday) (NT, n = 7; HCQ, n = 8). **f** Tumor volumes and weights measured after co-injection of EO771 and adipocytes. Adipocytes were pretreated with PBS or HCQ (40 μM) for 2 days. 2.5 × 10⁵ number of cancer cells and adipocytes were mixed to be subcutaneously injected into 7-week-old mice (Ctrl n = 7; HCQ n = 8). Statistics were calculated using ordinary two-way ANOVA (**c**, **d**) and two-tailed unpaired Students $t$-test (**c**, **d**, **e**, **f**). Western blot results are representatives of at least three independent experiments.

growth (Fig. 7d). Curtailed VP treatment still led to a regression of the tumor in BaKO, highlighting the potential impact of VP on tumors growing in adipose-rich environments (Fig. 7d). As diet-induced obesity mice develop various metabolic disorders involving inflammatory adipose tissue, we examined the relationship between BECN1 and YAP expression in the iWAT of HFD-fed mice. Interestingly, BECN1 levels were decreased while YAP levels were increased in the HFD-fed iWAT (Fig. 7e; Supplementary Fig. 9f). We further characterized HFD-fed iWAT and realized the overlapping properties between BaKO and HFD-fed iWATs (Supplementary Fig. 9g).

Therefore, we hypothesized that VP could drive tumor regression in mice with metabolic dysregulation and evaluated the impact of VP treatment in HFD-fed mice. While EO771 grew more rapidly in HFD-fed mice, VP treatment led to tumor regression only in HFD-fed mice (Fig. 7f, g; Supplementary Fig. 9h, i). In summary, these findings show that BECN1 and YAP/TAZ levels serve as the indicators of adipose tissue health and its potential to provide a pro-tumorigenic environment (Fig. 7h). Moreover, our findings advocate the use of YAP/TAZ inhibitors such as VP to prevent CAA transformation in the context of TME.

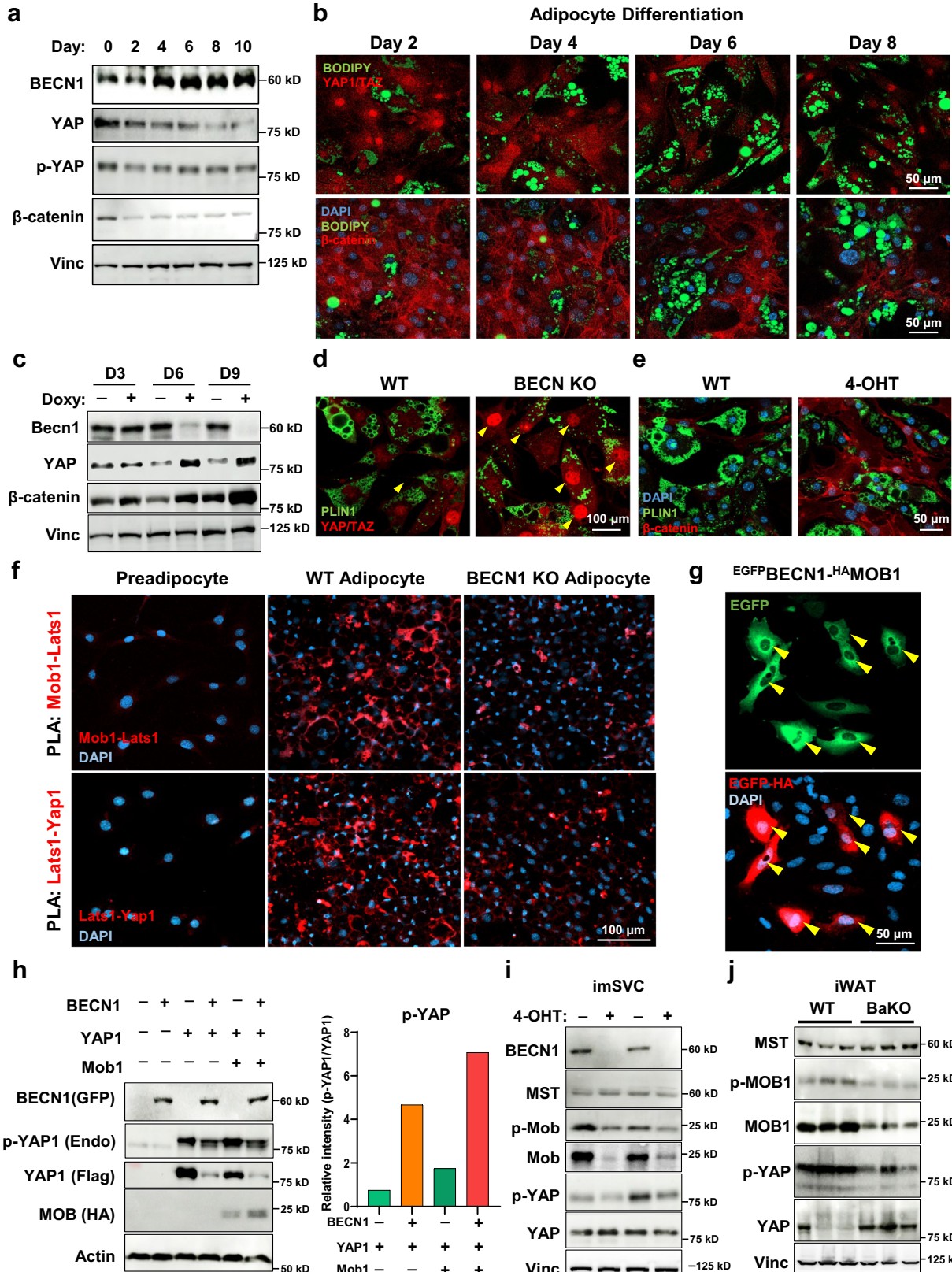

## Discussion

It is broadly accepted that CAAs undergo dedifferentiation, a complex process characterized by a change in morphology, downregulation of mature adipocyte markers, loss of lipid contents, and acquisition of mesenchymal stem cell phenotype[32,41–44]. These transformations of adipocytes could be critical to patients with advanced tumors since they lead to cancer-associated cachexia[45]. While only Wnt and Notch signaling have been identified as the mediators of adipocyte dedifferentiation for CAA development[32,44], we show that similar phenotypes, such as suppression of adipogenic markers and acquisition of fibroblast-like features, can be obtained from BECN1 depletion. Further, our transcriptomic analyses demonstrate that YAP/TAZ activity

**Fig. 5 | BECN1-deficient adipocytes suppress adipogenesis while inducing YAP/TAZ signaling. a** Western blotting analysis of protein lysates isolated from differentiating adipocytes over the time course. **b** Representative IF images throughout 10T1/2 adipocyte differentiation. Image was taken from 2nd day to 8th day of differentiation. Adipocytes were stained with YAP/TAZ (red, upper panel) and β-catenin (red, lower panel), BODIPY (green), and DAPI (blue). **c** Western blotting analysis of protein lysates isolated from doxycycline-inducible *Becn1 KO* cell lines. Fully differentiated adipocytes with and without doxycycline (Doxy) treatment for the time indicated. **d, e** Representative IF images of mature adipocytes stained with YAP/TAZ and (e) β-catenin. Arrowheads indicate nuclear YAP/TAZ. **f** Representative IF images of PLA conducted on preadipocytes, adipocytes, and BECN1 KO adipocytes. Interaction between endogenous MOB1-LATS1 and LATS1-YAP1 were detected. **g** Representative IF image of PLA conducted on U2OS cells overexpressed with BECN1[GFP] and MOB1[HA]. **h** Western blotting analysis of protein lysates isolated from HEK293 cells overexpressed with BECN1[GFP], YAP1[FLAG], and MOB1[HA] as indicated. The tag-antibodies were used to detect BECN1, YAP1, and MOB1. Endogenous p-YAP levels relative to YAP1 were quantified using imageJ. **i** Western blotting analysis of protein lysates isolated from imSVCs with and without 4-OHT treatment. **j** Western blotting analysis of protein lysates isolated from WT and BaKO inguinal white adipose tissues (iWATs). Mice were sacrificed at 8-weeks-old of age. Western blot results are representatives of at least three independent experiments.

modulates the dedifferentiation of adipocytes which are capable of accelerating tumor malignancy. This implies that the downregulation of BECN1 and activation of YAP/TAZ can trigger CAA transformation. Additionally, accelerated tumor progression observed in BaKO mice demonstrates that rapid development of CAAs, or preconditioned TME, provides a favorable niche for tumor growth.

Of note, BaKO WAT displays lipodystrophy phenotypes, such as shrinking adipocytes and lipid contents, which significantly differ from obese WAT. However, both conditions were suitable for rapid tumor progression and were sensitive to VP treatment. We identified multiple features of BaKO WAT that are analogous to those of obese WAT (Supplementary Fig. 9e). This highlights that instead of fat mass, the quality of adipose tissue, pertaining to inflammation, vascularity, adipokine secretion, and extracellular matrix integrity, is more crucial when evaluating the TME status. Furthermore, the WATs of the HFD-fed mice had decreased levels of BECN1 and increased levels of YAP/TAZ (Fig. 7e), suggesting that they developed impaired adipocyte conditions. While the relationship between obesity, diabetes, and cancer remains complex and multifactorial, mounting evidence has indicated a primary role of inflammatory factors and adipokines on tumor progression[23]. However, further studies on the adipocyte-derived factors are warranted to understand the contribution of obesity and diabetes in malignant TME formation, which will provide more effective pharmacologic and surgical interventions against obesity/diabetes-associated cancers.

Numerous studies have shown that autophagy-associated proteins may have functions that extend beyond autophagy. For example, the tumor suppressive role of UVRAG and BECN1 could be attributed to the trafficking function of BECN1/UVRAG in initiating autophagy[46], which regulates the cell surface localization of E-cadherin. This, in turn, prevents tumor progression via contact inhibition of proliferation through Hippo signaling, suppression of β-catenin, and loss of mesenchymal phenotype[46]. Thus, these properties of BECN1 could contribute to the adipocyte transformation observed in BECN1-deficient adipocytes and BaKO. In addition to its trafficking function, BECN1 directly interacts with various proteins, including Hippo-associated proteins, such as LATS1 and MST1[47,48]. These findings indicated a reciprocal interaction between two vital biological processes: autophagy and the Hippo pathway. However, the PLA results presented MOB1 as another candidate that interacts with BECN1. Although the mechanism through which BECN1-MOB1 proximity regulates MOB1 remains unidentified, we posit that the BECN1-MOB1-YAP/TAZ axis is pivotal in determining adipocyte differentiation state.

Our transcriptomic analysis on BaKO and peritumoral WATs well-characterized the dysfunctional adipocytes and identified IL-6-JAK-STAT3 signaling as the 5th and 9th most significantly upregulated gene set (Figs. 2a, 3f) in BaKO and peritumoral WATs, respectively. Previous studies have shown that, IL-6 expression is not only relevant to the pro-inflammatory signals released from impaired adipose tissue, but also relevant to the induction of Wnt/β-catenin activity through JAK2 and FOXO3[49,50]. In this study, we showed the upregulation of IL-6 in BaKO WAT while it was restored in BYTaKO WAT (Fig. 6f). Additionally, the

CHIP-seq results of YAP and TEAD indicate potential binding regions at the IL-6 promoter, suggesting that the activation of YAP directly regulates IL-6 expression (Supplementary Fig. 10a). Heatmap analysis on Wnt/β-catenin signaling in WT, BaKO, and BYTaKO WATs revealed a significant activation in BaKO but restoration in BYTaKO (Supplementary Fig. 10b). This trend aligns well with our study, wherein lipodystrophy phenotypes and tumor progression were restored in BYTaKO (Figs. 6 and 7). These findings suggest the IL6-JAK-Wnt/β-catenin axis as a plausible underlying mechanism for adipocyte transformation; however, further investigations are needed to validate this finding.

Previous studies have extensively investigated the role of the Hippo pathway in cancer cells, but our results highlight its importance in adipocytes as a component of the TME. Specifically, TAZ has been identified as a repressor of PPARγ, and TAZ-deficient adipocytes have exhibited improved health[51]. Additionally, research on TAZ in adipocytes has manifested its role in regulating resistin secretion and promoting tumorigenesis in triple-negative breast cancers[52]. Our study also reveals the significance of YAP/TAZ in adipocyte differentiation and homeostasis. Aberrant expression and activity of these proteins could mediate adipocyte transformation, which inflicts CAA-like effects on the TME. Moreover, the transformation was a preventable process since the suppression of YAP/TAZ signaling could maintain adipocyte homeostasis. This may be due to the YAP/TAZ function to modulate cellular plasticity and stemness, which are the critical determinants of cellular fate in multiple TME compartments[53]. Furthermore, we identified that YAP-TEAD could directly promote *LCN2* transcription (Fig. 6i–k). Thus, YAP/TAZ inhibitors, such as VP, may be particularly potent as they can target both intrinsic and extrinsic factors of cancer cells, leveraging the unique properties of YAP/TAZ for a dual therapeutic benefit.

To extend the scope of our study, we sought to explore the influence of the BECN1-YAP/TAZ axis in adipocytes on other components within the TME. Specifically, we investigated the impact of BECN1 loss in fibroblasts on cancer malignancy, presenting a potential avenue for further research. We co-cultured mouse embryonic fibroblasts (MEFs) and preadipocytes with EO771 and observed that with BECN1 depletion in fibroblasts, EO771 cells exhibited malignant behavior characterized by increased growth rate and EMT (Supplementary Fig. 11a–d). These findings suggest the contribution of the BECN1-YAP/TAZ axis to various components of the TME; however, further comprehensive research is necessary to validate the role of the BECN1-YAP/TAZ axis in fibroblasts. In summary, our findings underscore the essential factors in CAA transformation and their significance in breast and colon cancer models. Moreover, our study provides a viable approach to aim adipose-TME, particularly for patients with metabolic dysregulation such as obesity and diabetes.

## Methods
### Animals
Adipocyte-specific Becn1 deficient mice (BaKO) were generated as described previously[20]. Mice with floxed alleles of *Yap1* (Yap1[tm1a(KOMP)Mbp]),

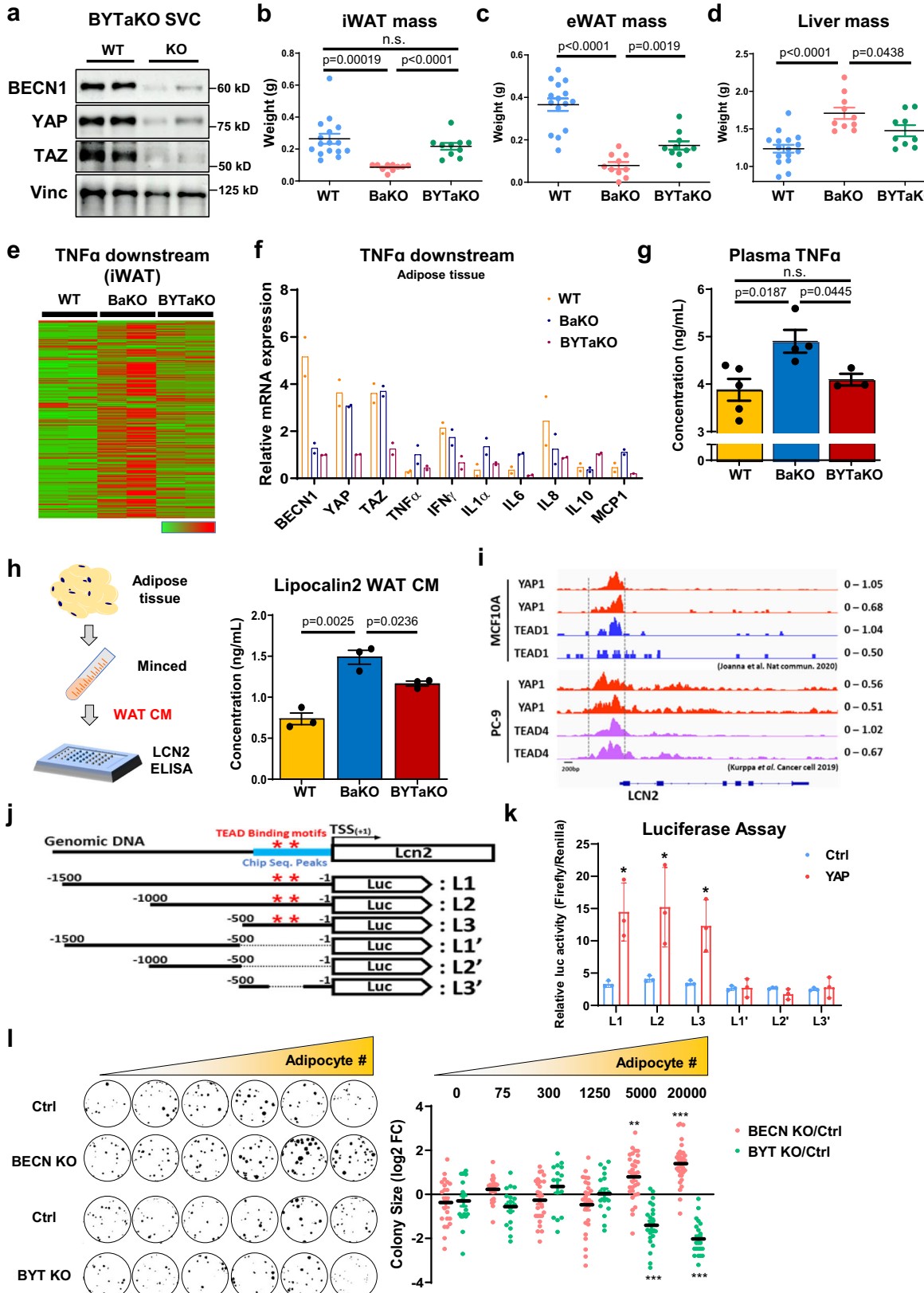

*Taz* (Wwtr1[tm1.2Eno]), and *Atg7* (Atg7[tm1Tchi] RBRC02759; RIKEN) were obtained from the Knockout Mouse Project (KOMP) Repository (KOMP, Davis, California, USA). BaKO mice were crossed with *Yap1*[flox/flox] (Yap1[tm1a(KOMP)Mbp]) and *Taz*[flox/flox] (Wwtr1[tm1.2Eno]) to generate adipocyte-specific *Becn1/Yap/Taz* KO mice (BYTaKO). Adipocyte-specific *Atg7* deficient mice were generated by crossing Adipoq-Cre (JAX: 010803)

with *Atg7*[flox/flox] (Atg7[tm1Tchi]). The excised alleles were validated using PCR analysis of genomic DNA extracted from the tail tips of mice. The genotyping primer sequences are listed in Supplementary Table 1. Adipoq-Cre, *Becn1*[flox/flox], MMTV-PyMT (PyBA) mice were generated by crossing BaKO with MMTV-PyMT (The Jackson Laboratory, Bar Harbor, ME, USA). The integrated MMTV-PyMT sequence was validated using PCR analysis

**Fig. 6 | BaKO phenotypes are restored in BECN1/YAP/TAZ triple KO mice.**
**a** Western blot analysis of protein lysates isolated from fully differentiated WT and adipocyte specific BECN1/YAP/TAZ KO (BYTaKO) SVCs. **b**–**d** Organ weights measured from 8-week-old WT, BaKO, and BYTaKO (WT, n = 16; BaKO, n = 10; BYTaKO, n = 10). **e** Heatmap analysis of TNFα downstream gene expression obtained from WT, BaKO, and BYTaKO iWAT. **f** Relative mRNA expression of BECN1, YAP, TAZ, and TNFα downstream genes confirmed by RT-qPCR (n = 2 mice per genotype).
**g** Concentration of TNFα in WT, BaKO, BYTaKO mouse serum using ELISA (WT, n = 5; BaKO, n = 4; BYTaKO, n = 3). **h** Concentration of LCN2 in WT, BaKO, BYTaKO iWAT CM using ELISA. iWATs were resected and minced to obtain WAT CM (n = 3 mice per genotype). **i** Publicly available YAP1, TEAD1, and TEAD4 CHIP-seq reads around LCN2 gene loci. CHIP-seq was conducted in MCF10A and PC-9 cells. **j** Schematic representation of luciferase reporter construct containing LCN2 promoter region. Red stars indicate the TEAD binding motifs, and blue bar indicates the regions with high CHIP-seq reads. **k** Dual Luciferase assay conducted in HEK293 cells transfected with the luciferase reporter constructs (LCN2). After transfection, recombinant YAP1 protein was treated (1 µg, 48 h) (n = 3, biologically independent samples per group). **l** Colony assay of EO771 co-cultured with BECN1 or BECN1/YAP/TAZ KO adipocytes. A total of 30 cancer cells were seeded with differentiated adipocytes, with adipocyte numbers indicated. Sizes for each EO771 colony were quantified using ImageJ and presented relative to the control, expressed as log2-fold change (FC) (30 distinct colonies per group). This experiment was conducted once. Statistics were calculated using two-tailed unpaired Students t-test (**b**, **c**, **d**, **g**, **h**, **l**). Data are shown as mean ± SEM; *$p \leq 0.05$, **$p \leq 0.01$, ***$p \leq 0.001$. Western blot results are representatives of at least three independent experiments.

of genomic DNA extracted from tail tips of mice. Mice received either a normal chow diet (NCD) or high-fat diet (HFD) (consisting of 60 kcal %), in addition to water, *ad libitum*, for indicated times. Immunocompromised N2G (NOD; Prkdc$^{em1Gmcr}$; Il2rg$^{em1Gmcr}$) mice were obtained from Gemcro Corp (Seoul, Republic of Korea). These mice were generated from NOD with deletion of entire exons of *Prkdc* and *Il2rg*.

All mice were housed in pathogen-free facilities under a 12 h light/dark cycle with unrestricted access to food and water. All animal care and experiments were performed in accordance with the guidelines of the Korean Food and Drug Administration and approved by the Institutional Animal Care and Use Committees (IACUC) of the Laboratory Animal Research Center at Yonsei University (permit number IACUC-A-202208-1520-01). The 6-8 weeks old female mice were used for breast cancer (EO771, 4T1, MMTV-PyMT) related experiments, while 6–8 weeks old male mice were used for colon cancer (MC-38) related experiments. The age of each mouse cohort is indicated in the figure legends, and their weights ranged from 20 to 30 g. The maximal tumor burden (10% of the mouse body mass) was permitted by IACUC, and all experiments involving mouse xenografts were terminated as soon as the tumor size reaches the maximal tumor volume.

### Adipose tissue fraction
White adipose tissues (WATs) were isolated from mice, dissected out, chopped, and incubated in collagenase buffer for 45 min at 37 °C with shaking. Samples were centrifuged to separate adipocytes (supernatant) and stromal vascular fractions (pellet). Stromal vascular fractions were seeded on a cultured plate to obtain stromal vascular cells (SVC). Peritumoral adipose tissues were obtained two weeks after injection of $1 \times 10^6$ EO771 cells at the region within 5 mm of the primary tumor.

### Cell cultures and adipocyte differentiation
The 3T3L1 (#CL-173), 10T1/2 (#CCL-226), EO771 (#CRL-3461), 4T1 (#CRL-2539), B16F10 (#CRL-6475), LL/2 (#CRL-1642), TC-1 (#CRL-2493), U2OS (#HTB-96) and HEK-293T (#CRL-11268) cell lines were purchased from American Type Culture Collection (ATCC). The MC-38 (#SCC172) murine colon adenocarcinoma cell line was purchased from Sigma-Aldrich. Cells were maintained in DMEM (Gibco) supplemented with 10% fetal bovine serum (FBS, Gibco) and 1% pen/strep (Gibco), and incubated at 37 °C in 5% $CO_2$.

For co-culture experiments, differentiated adipocytes and EO771 murine mammary cancer cell line were cultured together in 60 pi (SPL #209260) or 6-well (SPL #37006) co-culture chambers. Colony formation assays were performed by mixing various numbers of differentiated adipocytes (incubated for 2 days in differentiation media and then 4 days with DMEM, 1 µM insulin (I9278, Sigma-Aldrich), 10% FBS, and 1% pen/strep with the EO771. After staining the cells with crystal violet (548-62-9, Sigma-Aldrich), colony sizes were measured using ImageJ 1.53t (National Institutes of Health). To prepare the conditioned media (CM), differentiated adipocytes were incubated for 48 h in a medium consisting of DMEM, 1% FBS, and 1% pen/strep. The CM was then 5-fold concentrated using a centrifugal filter (Merk, C7715) and mixed with DMEM, 3% FBS, and 1% pen/strep in a 1:1 ratio before treatment. Cell viability was measured using the MTT assay (M6494; Invitrogen).

### Syngeneic and xenograft models
Approximately $5 \times 10^5$ MC-38 and EO771 cells were injected into the subcutaneous or mammary fat pads of WT, BaKO, or BYTaKO mice, and tumor sizes were periodically measured using the following equation.

$$volume = \frac{1}{2}(length \times width^2) \qquad (1)$$

For orthotopic injection of mouse colon cancer cells, $5 \times 10^5$ number of MC-38 was injected into mouse cecum. The cells were injected at the apex region of the cecum. After 4 weeks of tumor progression, the mice were sacrificed, and colons were resected. Tumors were carefully excised from the cecum and weighed to determine the final tumor mass. Mice aged 7-8 weeks were used for this study. Mice were euthanized by cervical dislocation, and tumor weights were measured after resection. WAT depots and metastasized lungs and livers were dissected for further assessments. Fresh tumor and WATs were immediately collected and kept on dry liquid nitrogen for RNA and protein extraction. Fresh tissues were immediately stored at −80°C for further use.

4T1 cells and differentiated imSVCs were mixed in a 1:1 ratio for adipocyte–cancer co-injection. The imSVCs were pre-treated with differentiation media for 2 days. After 4 days, 4-hydroxytamoxifen (4-OHT) was treated (500 nM, 48 h) to deplete BECN1. $2 \times 10^5$ 4T1 cells and varying number of adipocytes (between $1 \times 10^4$ and $2 \times 10^5$) were mixed in PBS. The mixture was then injected subcutaneously into N2G mice.

### Immortalization of SVCs and conditional BECN1 KO adipocytes
Human embryonic kidney 293 (HEK293) cells were transfected with simian virus 40 large T antigen containing plasmid using Lipofectamine. Media was then harvested 40 h post-transfection. The virus-containing media in the presence of polybrene (5 µg/mL) was treated to SVCs extracted from mice mated with *Becn1*$^{f/f}$ and ROSA-Cre$^{ERT2}$. After 8 h of incubation, the viral supernatant was removed and reinfected for another day. Each colony of the immortalized SVCs were tested and selected based on the adipocyte differentiation efficiency. Once terminally differentiated, imSVCs were treated with tamoxifen (500 nM, 72 h) to deplete BECN1[20].

### Proximity ligation assay (PLA)
U2OS cells were transfected with plasmids to express our proteins of interest. Cells were washed with PBS and fixed in 4% paraformaldehyde for 15 min at room temperature. After fixation, cells were

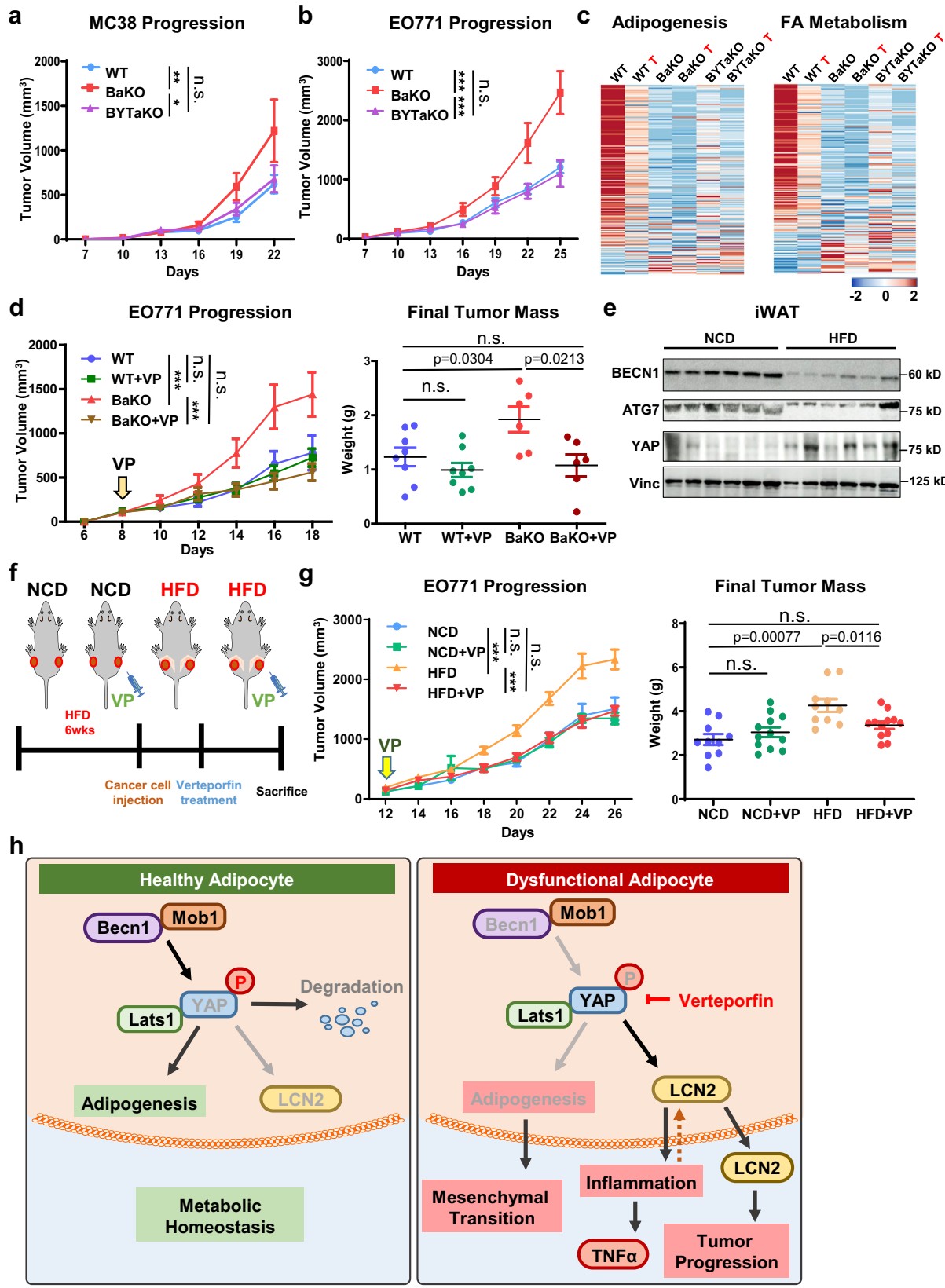

Nature Communications | (2024)15:4052                                                                                    13

permeabilized with 0.1% Triton X-100 for 10 min and blocked in the blocking solution for 1 h. Primary antibodies specific to the proteins of interest or tags were added to the cells and incubated overnight at 4 °C. After washing with PBS, PLA probes were added, and the cells were incubated for 1 h. Ligation and amplification steps were then performed. Following the PLA, cells were washed and counterstained with DAPI for nuclear visualization. The intensity of PLA signals was normalized by the cell number, using ImageJ.

**Dual-luciferase reporter assay**
HEK293 cells were seeded onto 12-well plates and allowed to grow overnight. Cells were transfected with luciferase reporter plasmid

**Fig. 7 | Inhibition of YAP/TAZ activity suppresses the tumorigenic effect of BECN1-deficient adipocytes. a** Tumor volumes measured after MC-38 subcutaneously injected into 7-week-old WT, BaKO, and BYTaKO mice (WT n = 20; BaKO n = 12; BYTaKO n = 12). **b** Tumor volumes measured after EO771 subcutaneously injected into 7-week-old WT, BaKO, and BYTaKO (WT n = 16; BaKO n = 6; BYTaKO n = 8). **c** Heatmap analysis of adipogenesis and FA metabolism gene sets between WT, BaKO, and BYTaKO iWATs and their peritumoral WATs (labeled with red letter 'T') (n = 2 each). **d** Tumor volumes and weights after mammary fat pad injection of EO771 into 7- week-old mice treated with or without VP (30 mg/kg, injected every other day). VP treatment was initiated when tumors reached a volume of 150 mm³, and the mice were sacrificed on day 22 following injection (WT, n = 8; WT VP, n = 8; KO, n = 6; KO VP, n = 6). **e** Western blot analysis of protein lysates isolated from iWAT of Normal Chow Diet (NCD) and High Fat Diet (HFD)-fed mice. Mice were fed with HFD chow for 12 weeks. **f** Schematic timeline of NCD and HFD-fed mice treated with or without VP. **g** Tumor volumes and weights after mammary fat pad injection of EO771 into NCD or HFD-fed mice. Verteporfin(VP) treatment (30 mg/kg, injected every other day) was initiated when tumors reached a volume of 150 mm³, and the mice were sacrificed on day 26 following the injection (n = 12, each group). **h** Schematic illustration depicting the signaling mechanism identified in this study. Statistics were calculated using ordinary two-way ANOVA (**a**, **b**, **d**, **g**) and two-tailed unpaired students t-test (**d**, **g**). Data are shown as mean ± SEM; *$p \le 0.05$, **$p \le 0.01$, ***$p \le 0.001$. Western blot results are representatives of at least three independent experiments.

constructs (pGL3) containing *LCN2* or *TNFα* promoter regions. Subsequently, YAP recombinant protein was added (1 μg). After 48 h, luciferase assay was conducted with the Dual Reporter Luciferase Assay System (E1910, Promega), according to the manufacturer's instruction. The relative levels of luciferase activity were normalized to the levels of Renilla luciferase activity.

### Retroviral shRNA design and transduction
The lentivirus inducing shBECN1 was generated through transfection with psPAX2, pmD2G, and pLV-H1TetO-GFP-Puro using Lipofectamine 3000 (Thermo Fisher # L3000001) following the manufacturer's instructions. For retroviral transduction of 10T1/2 (ATCC), polybrene (Merk #TR-1003-G) was used. The transduced cells were selected by incubation in medium containing DMEM, 1 μM insulin, 10% FBS, 1% pen/strep, and 2 μg/mL puromycin. The shRNA oligonucleotides targeting mouse BECN1, YAP, and TAZ were designed using the web service (https://biosettia.com/support/shrna-designer/) and their sequences are listed in Supplementary Table 2.

### RNA interference-mediated knockdown
The transfection of siATG7 (SC-29797; Santa Cruz), siBECN1 (SC-41447; Santa Cruz), and siLCN2R was carried out using Lipofectamine 3000 (Thermo Fisher # L3000001), following the manufacturer's instructions. siLCN2R was designed using the web service (https://biosettia.com/support/shrna-designer/) and synthesized by GenePharma. siLCN2R mixture was prepared by combining 2.5 ng/μL of each siRNA to yield a final concentration of 10 ng/μL. siLCN2R sequences are listed in Supplementary Table 3.

### Drug treatment
Drug uptake was assessed by administering 10 mg/mL Verteporfin (129497-78-5, USP) dissolved in 5% dimethyl sulfoxide (DMSO) corn oil (30 mg/kg or 50 mg/kg) via intraperitoneal injection in WT and BaKO mice.

### Quantification of free fatty acids, adipokines, and cytokines
The concentration of FFAs in the adipocyte-CM was quantified using an ELISA assay kit (ab65341; Abcam). Plasma adipokine levels in WT and BaKO mice were measured using the Mouse Adipokine Array Kit (ARY013; R&D Systems) and LEGENDplex (740929; Biolegend). Blood was collected from the retroorbital sinus using a capillary tube.

### Western blotting analyses and antibodies
Cells were harvested and lysed in RIPA buffer on ice for 15–30 min or lysed in SDS lysis buffer prior to sonication. Protein content was measured using Bicinchoninic acid (BCA) assay. Equal amounts of protein were separated using sodium dodecyl sulfate gel electrophoresis (SDS-PAGE) and transferred to a nitrocellulose membrane. The membranes were blocked with 5% skim milk in Tris-buffered saline with Tween 20 (TBST) and incubated with primary antibodies overnight at 4 °C, followed by HRP-conjugated secondary antibodies. Bands were visualized using enhanced chemiluminescence and

quantified using ImageJ. Western blotting was performed with the following antibodies diluted within 1:500 and 1:2000 range.: anti-BECN1 (#3738; Cell Signaling Technology & sc-48341; Santa Cruz Biotechnology), anti-FASN (#3180; Cell Signaling Technology), anti-PPARγ (#2435; Cell Signaling Technology), anti-PLIN1 (#9349; Cell Signaling Technology), anti-actin (sc-47778; Santa Cruz Biotechnology), anti-GAPDH (sc-32233; Santa Cruz Biotechnology), anti-LC3 (L8918; Sigma-Aldrich), anti-p62 (H00008878-M01; Abnova), anti-tubulin (sc-48341; Santa Cruz Biotechnology), anti-β-catenin (sc-7963; Santa Cruz Biotechnology), anti-YAP/TAZ (sc-101199; Santa Cruz), anti-YAP (#14074; Cell Signaling Technology), anti-TAZ (#4883; Cell Signaling Technology), anti-phospho-YAP (#13008; Cell Signaling Technology), anti-LCN2 (AF1857; R&D Systems), anti-HSL (#4107; Cell Signaling Technology), anti-phospho-HSL (#4137; #4139, #45804; Cell Signaling Technology), anti-MOB1A (sc-393212; Santa Cruz Biotechnology), anti-phospho-MOB1 (#8699; Cell Signaling Technology), anti-MST1 (#3682; Cell Signaling Technology), Goat anti-Mouse IgG(H + L)-HRP (SA001; GenDEPOT), Goat anti-Rabbit IgG(H + L)-HRP (SA002; GenDEPOT), and Rabbit anti-Goat IgG(H + L)-HRP (SA007; GenDEPOT). Please see Source Data for all uncropped blots (Main and Supplementary Figs.).

### Immunofluorescence imaging microscopy
Before seeding, a cover glass (Marienfield) was coated with a 1% (w/v) gelatin solution in PBS and incubated for 30 min at 37 °C. After washing the cells with PBS, they were fixed with 10% formalin in PBS for 15 min at room temperature (RT). Following three washes with PBS, the cells were permeabilized with 0.5% Triton X-100 in PBS for 10 min at RT. After three additional washes with 0.1% Triton X-100 in PBS, the cells were blocked with 5% BSA and 0.5% Triton X-100 in PBS for 30 min at RT. The cover glass was placed on parafilm inside a humid chamber and incubated for 2 hr at RT with the following primary antibodies, which were diluted to 1:100 in 5% BSA and 0.5% Triton X-100 in PBS: anti-BECN1 (#3738; CST and sc-48341; Santa Cruz Biotechnology), anti-PLIN1 (#9349; CST), anti-β-catenin (sc-7963; Santa Cruz Biotechnology), anti-YAP/TAZ (sc-101199; Santa Cruz Biotechnology), anti-YAP (#14074; CST), and anti-TAZ (#4883; CST). After washing the cells three times with 0.1% Triton X-100 in PBS, 5 min per wash, they were incubated with secondary antibodies conjugated with fluorophores (A32744 and A21206; Invitrogen) diluted to 1:100 in PBS with 0.1% Triton X-100. Following three additional washes with 0.1% Triton X-100 in PBS for 5 min each, the cells were mounted on glass slides using Fluoroshield with 4′,6-diamidino-2-phenylindole (DAPI) (F6057; Sigma-Aldrich). Conventional fluorescent imaging was performed using the Axio Observer Z1/7 (Carl Zeiss), while confocal images were captured using LSM980 (Carl Zeiss).

### RNA extraction, reverse transcription quantitative real-time PCR, and RNA sequencing
Total RNA was extracted from the tissues and cells using TRIzol Reagent (#15596026; Invitrogen) and RNeasy Mini Kit (#74104:Qiagen, Hilden, Germany) under the manufacturer's instructions. First-strand cDNA was prepared from 500 ng of total RNA using RevertAid First

Strand cDNA Synthesis (#K1622; Thermo Fisher Scientific) according to the manufacturer's instructions. For quantitative PCR analysis, primers were designed and are listed in Supplementary Table 4. Expression of each gene was normalized to the housekeeping genes, *GAPDH* and *β-actin*.

For RNA-sequencing, adipose tissue samples were harvested from tumor injected WT, BaKO, BYTaKO mice. Total RNA was extracted using the RNeasy Mini Kit and ensured the RNA integrity number of the samples were above 9. RNA-seq was then carried out by Geninus (Seoul, South Korea). Gene expression levels were quantified and normalized as transcripts per million (TPM) to facilitate comparison across samples. Differential expression analysis was conducted using Gene Set Enrichment Analysis (GSEA) with statistical significance set at an normalized *p*-valued (NOM *p*-val) <0.05.

### Statistical analysis

No statistical methods were used to predetermine sample size. Sample sizes were estimated based on experiences on the similar experiments performed by us and other published studies. Data are presented as mean ± SEM. Statistical analysis was performed using GraphPad Prism 5 software (Graph-Pad Software, Inc.). To compare two groups, the unpaired Student *t*-test was used. For comparisons of more than two groups, a two-way ANOVA was employed. Repeated-measures ANOVA was used to analyze body weight and GTT data. The Bonferroni post hoc tests were used to determine significant differences after ANOVA. Statistical significance was considered at $p < 0.05$.

### Reporting summary

Further information on research design is available in the Nature Portfolio Reporting Summary linked to this article.

## Data availability

All data are available in the main text, supplementary materials or relevant repositories. RNA-seq data are uploaded in public database 'Genomic Sequence Archive (GSA)' with publicly available accession code CRA011844 (https://ngdc.cncb.ac.cn/gsa/search?searchTerm=CRA011844+). Source data are provided with this paper and deposited in Figshare (https://doi.org/10.6084/m9.figshare.25101185). Plasmids, cell lines, and other materials in this study can be shared upon request. For further information and requests for resources and reagents, please contact Han-Woong Lee (hwl@yonsei.ac.kr).

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

## Acknowledgements

This study was funded by National Research Foundation of Republic of Korea (NRF-2020M3F7A1094089) and Ministry of Science and ICT (RS-2023-00224201), National Cancer Center (HA22C0147), and Ministry of Education (2022R1A6A3A13071199). We acknowledge the support from Brain Korea 21 (BK21) Four program and Yonsei University.

## Author contributions

Y. Song: Data Curation, validation, investigation, visualization, metho-dology, writing-original draft and editing. H. Na: Data Curation, valida-tion, investigation, visualization, methodology, writing-original draft. S.E. Lee: Software, methodology. Y.M. Kim: Data curation. J. Moon: Data curation. T.W. Nam: Data curation. Y. Ji: Methodology, writing-review. Y. Jin: Methodology, investigation. J.H. Park: Data curation. S.C. Cho: Data Curation. J. Lee: Conceptualization, writing-review D. Hwang: Bioinfor-matics, conceptualization. J.B. Kim: Conceptualization, writing-review. S.-J. Ha: Conceptualization, resources. H.W. Park: Conceptualization, resources. H.-W. Lee: Conceptualization, resources, supervision, inves-tigation, writing-review, editing, project administration, and funding acquisition.

## Competing interests

The authors declare no competing interests.

## Additional information

**Peer review information** *Nature Communications* thanks Teresa Mon-kkonen, Kai Sun and the other anonymous reviewer(s) for their con-tribution to the peer review of this work. A peer review file is available.

