## [Peer Review File · Nature Communications]

Dysfunctional Adipocytes Promote Tumor Progression Through YAP/TAZ-dependent Cancer-Associated Adipocyte TransformationREVIEWER COMMENTS

Reviewer #1 (Remarks to the Author):

This manuscript reports a tumor suppressive role of BECN1 in control of cancer-associated adipocytes (CAAs) to inhibit cancer development in adipose-rich tissues. The authors show that the BECN1-deficient adipocytes promote breast and colon tumor growth, which is independent of BECN1-mediated autophagy but involved in the activation of Hippo pathway effectors YAP and TAZ. Upon BECN1 loss, activated YAP/TAZ drive the transcription of genes in inflammation, TNF-alpha signaling, and secreted adipokines to induce tumor progression, phenocopying the oncogenic events observed in CAAs. Targeting YAP/TAZ genetically and pharmacologically inhibits adipose-associated tumor growth.

Although the proposed model is potentially interesting, some of the results presented are too preliminary and require more robust data to support the authors' conclusions. Another weakness of this study is that the presented findings are mostly descriptive, lacking mechanistic insights into the observations.

The following improvements are suggested for the authors.

1. The xenograft tumor studies (Fig. 1a-d, 4e-g, 7) were performed subcutaneously, which lacks biological relevance for the related tumor microenvironment. Orthotopic xenograft should be performed. Additional cell lines should be included to address the potential cell line specificity issue.
2. It is unclear how YAP/TAZ were identified to act downstream of BECN1 in adipocytes. If this is the case, why was YAP/TAZ or Hippo signaling gene signature not identified as the top altered gene set in the BECN1-deficient cells (Fig. 2a, 3f)? How do YAP/TAZ regulate the transcription of inflammation genes as well as LCN2 (Fig. 7f-7i)? Are YAP/TAZ and TEAD able to directly bind the promoters/enhancers of these genes? ChIP assay should be performed to test this hypothesis. To strengthen this conclusion, the authors should also examine the adipocytes with Hippo deficiency (e.g., Lats1/2 KO) or YAP activation (e.g., YAP-5SA) to see whether they could similarly modulate the inflammation/LCN2 genes and tumor growth.
3. In Fig. 5a and 5c, YAP expression is significantly suppressed by BECN1. However, in Fig. 5h, loss of BECN1 did not affect YAP protein level. This inconsistency should be addressed. Would YAP transcription be affected in Fig. 5a, 5c, and 5h? This brings in another question – how does BECN1 regulate the Hippo pathway? Although the authors cited some literatures, they should at least examine additional Hippo pathway components like LATS1, MOB1, MST1 (both expression and phosphorylation) in the BECN1-deficient cells.
4. In Fig. 4, the authors show a dispensable role of autophagy in BECN1-mediated regulation of YAP/TAZ and tumor growth. To strengthen the conclusion, the authors should reconstitute the BECN1-deficient adipocytes with wild-type BECN1 and its mutants – the ones that cannot activate autophagy and the Hippo pathway, respectively, and use these cells to evaluate their roles in driving inflammation genes' transcription and tumor growth.
5. BECN1 protein expression in adipocytes were tightly controlled by cancer cells (Fig. 3c), adipocyte differentiation (Fig. 5a), and high-fat diet (Fig. 7e). Since these data provide the physiological/pathological regulation of BECN1 in adipocytes, the underlying mechanisms

should be elucidated.

6. The reviewer is curious about what if BECN1 is depleted in other types of cells like fibroblasts and epithelial cells. Would these BECN1-deficient non-adipocytes cells also show YAP/TAZ activation and enhanced inflammation, and drive tumor growth by modulating the microenvironment?

Minor points:

1. In Fig. 2j, the authors should knockdown LCN2 in adipocytes to repeat the co-culture experiment. In addition, treatment with LCN2 antibody in the co-culture media would help further strengthen the conclusion.

2. In Fig. 3c-d, were the q-PCR and WB studies performed in the mixed adipocyte-cancer cells or specifically in the isolated adipocytes from the co-culture system?

3. The provided Figure Legends are quite concise, making it hard to determine how the experiments were exactly performed.

Reviewer #2 (Remarks to the Author):

In this manuscript, the authors employed an adipose tissue-specific BECN1 knockout (BaKO) mouse model to investigate the effects of dysfunctional adipocytes (caused by loss-of-function of Beclin1) on the growth of certain types of tumors. They reported that BaKO adipocytes promote tumor growth both in vitro and in vivo using a combination of multiple mouse models. Furthermore, they demonstrated the involvement of the YAP/TAZ signaling pathway in tumor progression. As a demonstration, deletion or inhibition of YAP/TAZ in the BaKO adipocytes restored normal cell functions, suppressed lipodystrophy, and mitigated inflammatory phenotypes. Overall, the data are robust, and the findings are intriguing. The conclusions are well-supported by their results. However, there are significant concerns regarding unclear presentations and a lack of sufficient mechanistic studies, which somewhat diminish the enthusiasm for this otherwise interesting study. The following concerns, including both major and minor issues, are outlined below:

Major Concerns:

1). Several result titles start with "Dysfunctional adipocytes," which fails to provide precise information about the adipocytes in the respective paragraph. If these cells are BECN1 KO cells, it is recommended to use "BaKO adipocytes."; If the cells were isolated from diet-induced obese adipose tissue, then the term "diet-induced obese adipocytes" should be used. Employing clearer terminology would help readers better understand the nature of these cells.

2). It has been reported that loss-of-function of BECN1 promotes tumor growth in cancer cells. Similarly, here, the authors found that BaKO in adipocytes stimulates co-cultured tumor cells and activates downstream YAP/TAZ pathways, paralleling the role observed previously in cancer cells. The stimulation of the YAP/TAZ signaling pathway might explain the impact on adipogenesis reduction and inflammation enhancement. However, one significant effect of BaKO in adipocytes is enhanced lipolysis (Fig. 5g, Supplementary Fig. 5e and i). Considering that lipolysis can promote cancer growth, it's surprising that the authors didn't investigate the mechanism underlying how BaKO adipocytes increase

lipolysis.

3). The manuscript lacks crucial information about the recipient mice for co-injected WT or *Becn1* KO cells with 4T1. Are these nude mice with compromised inflammatory reactions for the injected cancer cells? Additionally, the nature of the co-injected fat cells remains unclear, as there is mention of imSVCs (line 99) and differentiated adipocytes (line 100).

4). There's a need to reconcile the observations of BECN1-deficient adipocytes and HFD-feeding induced obese adipocytes. In Fig 6, the authors note that inhibition of YAP/TAZ activity in both BECN1-deficient and HFD-fed obese adipocytes yields the same tumor suppression effect. However, these two adipocyte types exhibit stark morphological differences: BECN1-deficient adipocytes have small lipid droplets and are prone to dedifferentiation to fibroblasts, while HFD-induced obese adipocytes possess unusually large lipid contents and do not transform into fibroblasts. Moreover, there's confusion in line 298-299, where they state that "the WATs of the HFD-fed mice had decreased levels of BECN1," while these fat cells do not share the lipid droplet phenotype of BECN1 KO adipocytes. Addressing these discrepancies is crucial.

5). The autophagy-independent effect of BECN1 in adipose tissue needs further experimentation to provide solid support. While the authors discuss the involvement of key factors such as LATS1 and MST1 (lines 303 to 312), more experiments are required to establish their role in regulation.

6). The discussion lacks depth and detail. The authors should provide more comprehensive mechanistic insights into the effects of BECN1 in adipocytes. Particularly, they should elucidate how BECN1 suppresses the YAP/TAZ signaling pathway specifically in adipocytes and compare their findings about BECN1/YAP/TAZ signaling in adipocytes with previous reports in other cell types.

Minor Concerns:

1). The RNA-seq results (Fig. 2a&b) were obtained from iWAT, which includes multiple cell populations. However, BECN1 deficiency is specific to adipocytes in the isolated fat tissue.

2). In line 165, clarify whether "Co-regulated" means "Co-upregulated" or "Co-downregulated."

3). In line 219, clarify what "restored phenotypes" refer to. Also, given that the KO effects are specific to adipocytes, it's important to clarify that the liver phenotypes aren't a direct result of YAP/TAZ KO.

4). Line 279: Clarify the specific KO mouse models used in the study.

5). In line 290, specify which phenotypes are meant in relation to the effects of BECN1 depletion.

6). In line 298, explain what is meant by the "quality of adipose tissue." Does "quality" refer to the capacity for lipolysis, appropriate adipokine secretion, or another aspect?

Reviewer #3 (Remarks to the Author):

Summary:

This paper demonstrates *Becn1* conditional KO in adipocytes can promote tumor growth of breast cancer and colorectal cancer cells in vivo and in vitro. Adipokine array analysis of serum plasma from *Becn1* mutant mice, and RNA sequencing of white adipose tissue of mutant and control mice revealed upregulation of LCN2 and TNF α signaling in mutants in concordance with the tumor growth phenotype. Further analysis suggests that BECN1 KO in adipocytes creates a pro-tumorigenic environment by eliciting gene transcription and

markers associated with CAA (cancer associated adipocyte) status. Upregulation of YAP/TAZ gene signatures was observed in both BECN1 KO and peritumoral adipocytes, suggesting a potential mechanism for the tumor growth phenotypes observed. Beclin KO mice also lacking YAP and TAZ reversed many lipodystrophy phenotypes observed, as well as the increase in TNF α levels. Since Atg7 conditional KO did not elicit similar YAP/TAZ or tumor growth phenotypes in vitro or in vivo that were observed in BECN1 KO, the authors speculate that perhaps BECN1 function via PI3K/MTOR may be responsible for some of the phenotypes. It is my opinion that this work is novel and important, and suitable for publication in this journal with some revisions to characterize the phenotypes and bolster molecular data.

Major Comments:

1. It would be good to look at LCN2 (TNF α ?) in the context of ATG7/ Baf/HCQ in vitro.
2. Fig 1h, add p62 and/or lc3b to western blot, to help add context for the Atg7 data.
3. Along with changes in tumor cell proliferation, it would be good to evaluate changes in cell death, i.e. by CC3 staining (i.e. Fig 6), since autophagy can regulate cell death.
4. Please comment/discuss on YAP/TAZ inhibitor effect on other stromal cells? Could be relevant given this paper: The β -catenin/YAP signaling axis is a key regulator of melanoma-associated fibroblasts
5. Are there any changes in Wnt signaling given the beta catenin changes? Can you comment on IL6 given its previous role? Do you think IL6 changes could also drive phenotypes?
6. Is LCN2 or the receptor expressed in the tumor cells of interest? Consider providing some staining and/or citations. (same for TNF α / receptor?) Is there any other previous data regarding LCN2 in breast/colorectal cancer specifically?
7. Please mention whether obesity is predictive of poor outcome in breast and colon cancer respectively. Overall, the discussion of cancer is very general, some more specifics for breast and colon (as well as cancers where there was no difference!) would be useful.
8. EMT: all the data for this are RNA levels/ gene signatures. Please add some protein data as well for a few markers. Please discuss, are EMT signatures associated with CAA status?
9. Any thoughts as to why the 2 other tumor cell types did not show the same phenotypes?

Minor comments:

1. Spell out LCN2 fully the first time it appears in the text.
2. Line 151 use "co-culture" instead of co cultivation
3. Should mention/discuss and cite Beige Adipocyte Maintenance Is Regulated by Autophagy-Induced Mitochondrial Clearance paper
4. Please spell out what are imSVCs the first time it's mentioned since I'm not familiar with this acronym.
5. Supp fig 4a: please add quantification
6. Supp. Fig 5c could benefit from BOIPDY stain for ease of interpretation.
7. Fig. 6g: is WT versus Beclin/YT CKO significantly different? Please add this to the figure.

**Response to Reviewers**

We highly appreciate the constructive comments provided by the reviewers. We
acknowledge that these concerns are reasonable and should be addressed in our
manuscript (especially the mechanistic insights into YAP/TAZ regulation and its potential
effects on other stromal cells in the tumor microenvironment). In this regard, we have
included our thorough responses here in blue and revised the text in the manuscript in red.
By doing so, we could further solidify our work, leading to a substantial improvement in the
quality of the manuscript. Once again, we express our great thanks to the reviewers for their
invaluable time and effort.

**REVIEWER COMMENTS**

**Reviewer #1 (Remarks to the Author):**

This manuscript reports a tumor suppressive role of BECN1 in control of cancer-associated
adipocytes (CAAs) to inhibit cancer development in adipose-rich tissues. The authors show
that the BECN1-deficient adipocytes promote breast and colon tumor growth, which is
independent of BECN1-mediated autophagy but involved in the activation of Hippo pathway
effectors YAP and TAZ. Upon BECN1 loss, activated YAP/TAZ drive the transcription of
genes in inflammation, TNF-alpha signaling, and secreted adipokines to induce tumor
progression, phenocopying the oncogenic events observed in CAAs. Targeting YAP/TAZ
genetically and pharmacologically inhibits adipose-associated tumor growth.

Although the proposed model is potentially interesting, some of the results presented are
too preliminary and require more robust data to support the authors' conclusions. Another
weakness of this study is that the presented findings are mostly descriptive, lacking
mechanistic insights into the observations.

We appreciate the reviewer's valuable feedback, emphasizing the need for more robust data
and mechanistic insights to support our conclusions. In response, we have dedicated efforts
to enhance the clarity and depth of our manuscript, specifically delving into the sequential
mechanism for BECN1 to YAP/TAZ and LCN2, addressing the reviewer's concerns and
elevating the overall quality of our work.

The following improvements are suggested for the authors.

1. The xenograft tumor studies (Fig. 1a-d, 4e-g, 7) were performed subcutaneously, which
lacks biological relevance for the related tumor microenvironment. Orthotopic xenograft
should be performed. Additional cell lines should be included to address the potential cell
line specificity issue.

We appreciate the reviewer's thoughtful suggestions and apologize for any confusion in
presenting the methodology of our in-vivo studies (Main Fig. 1,4,7). Our cancer cell injection
data comprise both breast and colon cancer models (EO771, 4T1, and MC38), along with
the MMTV-PyMT mouse model. Here we provide a detailed explanation of how we
implemented these models in our study.

EO771 – Breast cancer cells were allografted by mammary fat pad injection.

4T1 – Breast cancer cells were co-injected subcutaneously with differentiated adipocytes.
 This approach aimed to minimize the influence of normal adipose tissue, focusing primarily
 on the impact of co-injected BECN1 WT or KO adipocytes.

MC38 – Colorectal cancer cells were allografted by subcutaneous injection.

MMTV-PyMT – Spontaneous breast cancer model mice were mated with BECN1 aKO mice.

It is important to note that, our breast cancer models were performed orthotopically, while
 the colon cancer model was not. To address the concerns raised, we performed orthotopic
 injection of MC38 into WT, BaKO, and BYTaKO to confirm the effect of dysfunctional
 adipocytes on tumor growth particularly in the context of tumor microenvironment (TME).
 MC38 cells were injected in the mouse cecum, and the tumors were resected after 4 weeks
 (**Revised Fig. 1a**). Consistently, MC38 grew much faster in BaKO colon, but not as much in
 that of BYTaKO (**Revised Fig. 1b, and c**). We hope this clarification addresses the
 reviewer’s concerns regarding the biological relevance of our xenograft tumor studies.

**Revised Fig. 1a-c** – (a) Colon orthotopic injection of MC38 into mouse cecum. Mice were
 sacrificed after 4 weeks post-injection. (b) Image of resected colons harboring tumors. (c)
 Final tumor mass measured (WT n = 8, BaKO n = 10, BYTaKO n = 4).

**The results have been incorporated into our Supplementary Figure 9.**

2. It is unclear how YAP/TAZ were identified to act downstream of BECN1 in adipocytes. If
 this is the case, why was YAP/TAZ or Hippo signaling gene signature not identified as the
 top altered gene set in the BECN1-deficient cells (Fig. 2a, 3f)? How do YAP/TAZ regulate
 the transcription of inflammation genes as well as LCN2 (Fig. 7f-7i)? Are YAP/TAZ and
 TEAD able to directly bind the promoters/enhancers of these genes? ChIP assay should be
 performed to test this hypothesis. To strengthen this conclusion, the authors should also
 examine the adipocytes with Hippo deficiency (e.g., Lats1/2 KO) or YAP activation (e.g.,
 YAP-5SA) to see whether they could similarly modulate the inflammation/LCN2 genes and
 tumor growth.

The reason why YAP/TAZ or Hippo signaling did not appear in the altered gene set is that
Main Fig. 2a and 3f are based on the Hallmark gene set enrichment analysis (HGSEA),
which only consists of 50 general gene sets. To address this, we browsed the 'C6:
oncogenic signature gene sets' in Molecular Signature Database (MSigDB). The result is
shown in Supplementary Fig. 3c and d, indicating that YAP conserved signature gene set
was at the 14th position among of the most significantly altered gene set.

We thank the reviewer to pinpoint the specific mechanism that remained vague in our initial
conclusion. We relied on literatures to support our hypothesis that links YAP/TAZ to the
upregulation of inflammatory gene expression. Furthermore, the reviewer's comment to
check TEAD binding motif at the promoters/enhancers of associated genes (including LCN2)
was very helpful [1]. We collected ChIP results of YAP and TEAD from PC-9 and MCF10A
cell lines in ChIP database [2, 3]. The following figure illustrates the potential binding region
of YAP and TEAD at the promoter regions of LCN2 and TNF α (**Revised Fig. 2a and b**).

**Revised Fig. 2a,b** – (a,b) ChIP-seq result of YAP and TEAD in MCF10A and PC-9 cells at
the genomic loci of LCN2 (a) and TNF α (b). ChIP-seq results were collected from 'chip-
atlas.org'. Intensified peaks indicate potential binding regions of the proteins.

To further validate the regulatory role of YAP/TAZ-TEAD activity on TNF α and LCN2
transcription, we constructed luciferase vectors containing TNF α or LCN2
promoters/enhancers (**Revised Fig. 2c and d**). After transfection of each plasmid, we
treated YAP recombinant protein to activate transcription factor efficacy. The result of our
dual luciferase assay showed only a mild increase from the TNF α constructs (**Revised Fig.**
**2e**). Unlike TNF α , LCN2 constructs containing the binding motifs of YAP and TEAD
exhibited enhanced luciferase activities by more than 140-fold compared to the control
(**Revised Fig. 2f**). Taken together, we show that adipocyte-LCN2 could be enhanced by
active YAP/TAZ-TEAD, which can be applicable to dysfunctional adipocytes that exhibit
YAP/TAZ activation.

Initially, our summary figure illustrated that TNF α contributes to LCN2 amplification in
dysfunctional adipocytes. This was supported by Supplementary Fig. 2c, where recombinant
TNF α protein was sufficient to increase LCN2 level in adipocytes. However, we now show
that Hippo pathway directly contributes to LCN2 production in adipocytes. Therefore, we
have revised our summary figure to better represent the direct regulation of LCN2 via

YAP/TAZ, emphasizing the contribution of Hippo pathway to LCN2 production in adipocytes
(Main Fig. 7h).

**Revised Fig.2c-f** – (c,d) Design of the luciferase reporter plasmids containing of TNF α (c)
and LCN2 (d) promoters/enhancers. Red stars indicate the DNA recognition sequence for
TEAD. Blue bar indicates the potential binding region according to ChIP-seq results in
**Revised Fig. 2a,b**. (e,f) Dual luciferase assay of the luciferase reporter constructs for both
LCN2 and TNF α . HEK293 cells were transfected with the indicated plasmids, then treated
with recombinant YAP protein (1 μ g, 48 hr).

**The results have been incorporated into our Main Figure 6 and Supplementary Figure 8.**

3. In Fig. 5a and 5c, YAP expression is significantly suppressed by BECN1. However, in Fig.
5h, loss of BECN1 did not affect YAP protein level. This inconsistency should be addressed.
Would YAP transcription be affected in Fig. 5a, 5c, and 5h? This brings in another question –
how does BECN1 regulate the Hippo pathway? Although the authors cited some literatures,
they should at least examine additional Hippo pathway components like LATS1, MOB1,
MST1 (both expression and phosphorylation) in the BECN1-deficient cells.

We appreciate the reviewer's keen observation regarding the apparent inconsistency in our
data. It is important to note that different cell lines were used in Fig. 5a, c and Fig.5h.
Specifically, in Fig. 5a, c, we used 10T1/2 cell line, which derived from mouse embryonic
fibroblasts and established as preadipocytes. However, in Fig. 5h, we used stromal vascular
cells (SVCs) extracted from inguinal white adipose tissue (iWAT) of an adult mouse. Given
the distinct origins at different developmental stages and distinct pluripotency of these two
cell lines, variations in results were anticipated. The 10T1/2 results show more dynamic
regulation of YAP protein, while the effect was more subtle in SVCs. These properties of
10T1/2 and SVCs were advantageous in identifying the mechanism of how BECN1 regulates
YAP.

Here, we organized multiple experimental designs to uncover the regulation of YAP/TAZ.

Our comprehensive investigation, incorporating both in-vitro and in-vivo results revealed that
BECN1 does not regulate YAP/TAZ at the transcriptional level (**Revised Fig. 3a; Main Fig.**
**6f**).

**Revised Fig. 3a** – Immortalized SVC (imSVC) was differentiated into mature adipocytes,
 then treated with 4-OHT to deplete BECN1 expression. Expression of BECN1, YAP1, and
 TAZ were evaluated using RT-qPCR.

As mentioned, we acknowledge the significance of comprehensively exploring the
 mechanical aspects of how BECN1 regulates YAP/TAZ through experimental investigations.
 Therefore, we conducted a proximity ligation assay (PLA) to check the interactions between
 BECN1 and Hippo-associated proteins. Since PLA reveals the proximity between two
 proteins, such interactions may impact their functional activity or protein stability [4]. We
 used the U2OS cell line to overexpress BECN1 along with Hippo-associated proteins: YAP1,
 LATS1, MST1, and MOB1. Intriguingly, we observed a remarkably strong interaction of BECN1
 with MOB1 compared to that with other Hippo-associated proteins (**Revised Fig. 3b-e**), and the
 interaction was higher than that between LATS1 and MOB1 (**Revised Fig. 3f**). Since MOB1
 accounts for LATS phosphorylation, which is necessary for its kinase activity, we thought the
 interaction between BECN1 and MOB1 would regulate the Hippo pathway in adipocytes [5].
 Therefore, we tested whether loss of BECN1 regulates LATS activity through MOB1 in
 adipocytes. The results showed that the loss of BECN1 in mature adipocytes could drive the
 reduction of MOB1-LATS interaction (**Revised Fig. 3g**), leading to inhibition of LATS-YAP
 interaction (**Revised Fig. 3h**).

Revised Fig. 3b-h – (b-e) Proximity ligation assay (PLA) performed in U2OS cells transfected with BECN1^{GFP} and other Hippo-associated proteins (MOB1, YAP1, MST1, and LATS1). Yellow arrows locate the interacting regions between the two proteins indicated. (f) LATS and MOB1 were transfected in U2OS as a positive control. (g-h) PLA was performed before and after the differentiation of 10T1/2 cells. Endogenous protein-protein interactions of MOB1-LATS1 and LATS1-YAP1 (h) were quantified.

To verify the interaction of BECN1-MOB1 causing Hippo pathway suppression, we tested
 the protein expression in both in-vitro and in-vivo system. imSVC result showed BECN1
 deletion did not alter the MST1 level, but it regulated YAP phosphorylation through the
 downregulation of MOB1 (**Revised Fig. 3i**). Consistently, mouse iWAT of BaKO contained
 similar levels of MST1 but low levels of MOB1 that leads to inhibition in YAP phosphorylation
 (**Revised Fig. 3j**). Next, we confirmed the effect of BECN1 on Hippo pathway via BECN1
 overexpression. We transfected BECN1, YAP1, and MOB1 as indicated and assessed the
 changes in YAP phosphorylation. Overexpression of BECN1 increased YAP phosphorylation
 level, which was more severely increased with the presence of MOB1 protein (**Revised Fig.**
 **3k**). Taken together, these findings suggest that BECN1 interacts with MOB1 to enhance
 Hippo pathway activity. Accordingly, we have included supporting results and revised our
 summary figure with more mechanistic insight (**Revised Fig. 3l**).

 **Revised Fig. 3i-l** – (i) Western blot analysis of imSVCs differentiated into mature
 adipocytes. 4-OHT was treated to deplete BECN1 after differentiation (j) Western blot
 analysis of protein samples isolated from WT or BaKO mouse iWATs (8 weeks old, n = 3).
 (k) Western blot analysis of HEK293 cells transfected with BECN1, YAP1, and MOB1 as
 indicated. Protein lysates were prepared 2 days post-transfection. (l) Revised summary
 figure.

**The results have been incorporated into our Main Figure 5 and Supplementary Figure 9.**

4. In Fig. 4, the authors show a dispensable role of autophagy in BECN1-mediated
 regulation of YAP/TAZ and tumor growth. To strengthen the conclusion, the authors should
 reconstitute the BECN1-deficient adipocytes with wild-type BECN1 and its mutants – the
 ones that cannot activate autophagy and the Hippo pathway, respectively, and use these
 cells to evaluate their roles in driving inflammation genes' transcription and tumor growth.

We agree that Main Fig. 4 should be more deliberately presented to emphasize the
 dispensable role of autophagy in BECN1-mediated regulation of YAP/TAZ. Unfortunately,
 the non-autophagic function of BECN1 has not been profoundly studied. Additionally, no
 specific domain of BECN1 that ablates its autophagic function *per se* has been identified.
 While mutation in the BH3 domain, which dissociates BECN1 from BCL2, is known to inhibit
 autophagy flux, its impact is not limited to autophagy and could affect other potential roles of
 BECN1 [6, 7]. Hence, our study is confined to testing of autophagy inhibition across different
 stages of the autophagic process, which may lead to alteration in the Hippo pathway.

To validate the effect of autophagy inhibition in adipocytes, we conducted RNA-sequencing
 on hydroxychloroquine (HCQ) treated adipocytes. Its effect on the Hippo pathway was rather
 opposite compared to what we observed in BECN1 deficient adipocytes (**Revised Fig. 4a**).
 Additionally, HCQ treatment in general reduced YAP/TAZ as well as β -Catenin throughout
 adipocyte differentiation (**Revised Fig. 4b**). Next, we evaluated how HCQ-treated adipocytes
 contribute to cancer growth. We pretreated HCQ on differentiated adipocytes and then co-
 cultured them with cancer cells (30 cells of EO771). After 10 days of co-cultivation of
 different number of adipocytes and cancer cells, the growth of EO771 was measured
 through crystal violet staining. The results revealed that cancer cells tended to form bigger
 colonies with an increased number of adipocytes; however, no remarkable difference was
 observed between the Ctrl and HCQ treated adipocytes (**Revised Fig. 4c**). Further, we
 tested this phenomenon in-vivo by co-injecting adipocytes and cancer cells. For this,
 adipocytes were pretreated with HCQ for 2 days and mixed with cancer cells and the mixture
 was injected subcutaneously into the mice. While EO771 co-injected with BECN1-deficient
 adipocytes enhanced tumor progression (Main Fig. 1i), HCQ-treated adipocytes did not
 (**Revised Fig. 4d**). Our revised data demonstrate that mere autophagy inhibition in
 adipocytes is not sufficient to regulate YAP/TAZ or promote tumor progression. This
 suggests that the distinctive function of BECN1 is responsible for driving alterations in the
 Hippo pathway.

**Revised Fig. 4a-d** – (a) Heat map analysis of YAP-conserved signature gene set (GSEA)
from RNA-seq results of Control versus HCQ-treated adipocytes. Fully differentiated
adipocytes were treated with HCQ (20 μ M, 48 hr). (b) Western blot analysis of Control and
HCQ treated (50 μ M, 48 hr) adipocytes at different stages of differentiation. Day 0, 6, and 10
represent different timepoints after differentiation stimulation. (c) Colony assay of EO771 co-
cultured with adipocytes treated with or without HCQ. HCQ was pretreated (20 μ M, 48 hr) to
the number of adipocytes as indicated. (d) Tumor progression evaluated by the rate of tumor
growth in volume and final mass. Mature adipocytes were treated with either PBS or HCQ
for 48 hr and then mixed with an equal number of EO771 cancer cells. Mice were sacrificed
20 days after injection.

**The results have been incorporated into our Main Figure 4 and Supplementary Figure 4.**

5. BECN1 protein expression in adipocytes were tightly controlled by cancer cells (Fig. 3c),
adipocyte differentiation (Fig. 5a), and high-fat diet (Fig. 7e). Since these data provide the
physiological/pathological regulation of BECN1 in adipocytes, the underlying mechanisms
should be elucidated.

We apologize for providing only descriptive response to this issue. Our study primarily
focuses on investigating the impact of dysfunctional adipocytes on tumor progression. We
have identified YAP/TAZ activity as a critical determinant of adipocyte dedifferentiation,
enhancing the formation of pro-tumorigenic environments. Here, BECN1-deficient
adipocytes are utilized as a dysfunctional/dedifferentiated adipocyte model. Consequently,
delving into what controls BECN1 expression may appear somewhat tangential to the
primary focus of our study. We hope the reviewer understands this matter.

We have reported the two circumstances that result in suppression of BECN1 expression in
adipocytes: adipocytes influenced by cancer cells and adipocytes from high-fat diet (HFD)
fed mice. To extract common traits that could potentially regulate BECN1 levels, we
compared the RNA-seq results of peritumoral WAT (Main Fig. 3f) and HFD-fed WAT. We
newly conducted bulk-RNA sequencing in iWAT of HFD-fed mice and compared it to that of
the normal chow diet (NCD)-fed mice. HGSEA was performed to classify the significantly
upregulated and downregulated gene sets and identified 7 upregulated and 3 downregulated
gene sets that were commonly observed in peritumoral WAT (**Revised Fig. 5a**). These
results indicated the potential signaling pathways that can regulate BECN1 in two different
circumstances. Furthermore, we conducted principal component analysis (PCA) between
RNA-seq results of Control, peritumoral, HFD Control, and HFD peritumoral WATs. PCA
depicts the relationships between those samples based on gene expression patterns, where
the distance between samples reflects how similar or different these samples are from one
another. The PCA result indicates that HFD-fed WAT and peritumoral WAT share somewhat
similar traits (**Revised Fig.5b**). In other words, WATs under two different circumstances may
acquire similar gene expression patterns.

**Revised Fig. 5a,b** – (a) HGSEA result of bulk RNA-seq conducted on NCD and HFD fed
 mouse iWATs. A positive NES score indicates that such gene sets were upregulated in
 HFD-fed iWAT. Red stars indicate that those gene sets were also altered in peritumoral
 WAT in a similar pattern. (b) PCA result of Naïve (N), Peritumoral (P), HFD Naïve (HFDN),
 and HFD peritumoral (HFDP) WAT RNA seq data. The distance between each dot reflects
 similarity in gene expression patten between samples.

Another aspect regarding BECN1 regulation is that its level is significantly altered depending
 on the adipocyte differentiation status (Main Fig. 5a, b). Furthermore, BECN1 depletion led
 to acquisition of dedifferentiation phenotypes through the expression of preadipocyte
 markers including YAP and β -Catenin (Main Fig. 5c-f). We tested whether similar
 phenomenon occurs in CAAs, which are well-known to undergo adipocyte dedifferentiation
 [8-11]. For this, we conducted immunocytochemistry (ICC) on YAP and β -Catenin to observe
 their dynamics upon adipocyte-cancer co-culture. Similar to BECN1-deficient adipocytes, co-
 cultured adipocytes showed increased nuclear YAP and cytoplasmic β -Catenin expression
 (**Revised Fig. 5c**). These results reaffirm that the BECN1 level relies on the differentiation
 status. Though it is insufficient to pinpoint a specific molecule or signaling pathway, we can
 conclude that BECN1 is regulated by adipocyte plasticity and differentiation stimuli.

**Revised Fig. 5c** – Immunofluorescence staining of YAP, β -Catenin, PLIN, and DAPI in
10T1/2 cells. Cells were prepared as preadipocytes, adipocytes, and CAAs. Yellow arrows
indicate the nuclear translocation of YAP.

**The results have been incorporated into our Supplementary Figure 5**

6. The reviewer is curious about what if BECN1 is depleted in other types of cells like
fibroblasts and epithelial cells. Would these BECN1-deficient non-adipocytes cells also show
YAP/TAZ activation and enhanced inflammation, and drive tumor growth by modulating the
microenvironment?

This comment helped us to extend the applicability of our study to various components
within the TME. Given the significance of cancer-associated fibroblast (CAF) as one of the
key TME component, we conducted pertinent experiments using mouse embryonic
fibroblasts (MEFs) and preadipocytes (10T1/2) [12]. By doing so, interesting results were
obtained: (1) BECN-YAP/TAZ regulation was continually observable in fibroblasts and (2)
BECN1-deficient fibroblasts were also sufficient to promote cancer growth.

As previously mentioned, BECN1 depletion suppressed Hippo signaling (**Revised Fig. 3b-i**).
PLA assay was again conducted in preadipocytes to observe the interaction between Hippo-
associated proteins. Similar to differentiated adipocytes, BECN1 depletion in preadipocytes
also diminished MOB1-LATS1 interaction (**Revised Fig. 6a**).

**Revised Fig. 6a** – PLA performed in WT and BECN1-deficient preadipocytes (10T1/2). The
interaction between endogenous MOB1 and LATS1 was evaluated. Longer exposure was
used compared to **Revised Fig. 3f**.

To assess the impact of BECN1-deficient fibroblasts on cancer growth, we co-cultured MEFs
and preadipocytes with EO771 cells. Upon BECN1 depletion, MEFs and preadipocytes were
co-cultured with 1000 cancer cells for 4 days, followed by crystal violet staining of the live
cells. Although no notable difference between WT and BECN1 knockdown (KD) fibroblasts
was observed initially (**Revised Fig. 6b**), morphological difference was observed at a
microscopic level (**Revised Fig. 6c**). Furthermore, measurement of circularity of the cells
was measured revealed that the cancer cells influenced by BECN1-deficient fibroblasts
exhibited more round-shaped morphologies (**Revised Fig. 6d**). To magnify the effect of
fibroblast co-culture, we lowered the number of cancer cells and extended the exposure to 8
306 days instead of 4. In fact, changes in the cancer cell morphologies were more apparent
(**Revised Fig. 6e**). Furthermore, we verified the impact of conditioned media (CM) extracted

from WT and BECN1-deficient fibroblasts on cancer growth. Consistently, cancer cells
 grown in BECN1-deficient fibroblast CM exhibited an enhanced growth rate compared to WT
 fibroblast CM (**Revised Fig. 6f**).

**Revised Fig. 6b-f** – (b) Schematic overview of fibroblast-cancer co-culture system. Cancer
 cells were stained with crystal violet after exposure to WT and BECN1-deficient fibroblasts.
 (c) Microscopic image of crystal violet stained cancer cells (d) Morphology of each cancer
 cell in (c) was evaluated for its circularity using *imageJ*. (e) Schematic overview of the
 co-culture system. The live image of a cancer cell colony displays distinct morphologies.

The morphological change in cancer cells signifies modifications in the cellular cytoskeleton,
 cell-cell adhesion molecules, and the acquisition of stemness characteristics. These
 alterations contribute to the transition of cancer cells, affecting their shape, migratory
 capacity, and invasive behavior, typically known as epithelial to mesenchymal transition

(EMT) [13, 14]. To evaluate the physiological implication driven by WT and BECN1-deficient
 preadipocytes, we observed changes in the EMT markers. In parallel to the enhanced
 circularity, cancer cells underwent EMT upon co-culture with BECN1-deficient preadipocytes
 **(Revised Fig. 6g,h)**. While our results are not sufficient to conclusively establish the
 BECN1-YAP/TAZ-LCN2 axis as the key modulator for this phenomenon, we can assert that
 the level of BECN1 in fibroblasts is equally critical in shaping a malignant TME. In summary,
 BECN1-deficient fibroblasts demonstrate the capability to induce cancer malignancy through
 enhanced proliferation and EMT.

**Revised Fig. 6g,h** - Immunofluorescence staining of β -Catenin, Vimentin, E-Cadherin, Slug,
 and DAPI in co-cultured EO771 cells. Cancer cells were co-cultured with WT and BECN1-
 deficient preadipocytes for two days.

**The results have been incorporated into our Supplementary Figure 11.**

Minor points:

1. In Fig. 2j, the authors should knockdown LCN2 in adipocytes to repeat the co-culture
 experiment. In addition, treatment with LCN2 antibody in the co-culture media would help
 further strengthen the conclusion.

Thank you for highlighting the need to complement our adipocyte-cancer co-culture
 experiment. We have addressed this by treating LCN2 inhibitor (ZINC00640089). The result
 indicates that EO771 growth was mitigated by LCN2 inhibitor especially when they were co-
 cultured with BECN1-deficient adipocytes **(Revised Fig. 7a)**.

**Revised Fig. 7a** – EO771 co-cultured with WT or BECN1 KO adipocytes for 7 days. LCN2
 inhibitor (ZINC00640089) at 5 μ M was administered bi-daily to inhibit BECN1 KO adipocyte-
 induced cancer cell growth during co-culture. Cell viability was assessed with crystal violet
 staining and quantified using *ImageJ*.

2. In Fig. 3c-d, were the q-PCR and WB studies performed in the mixed adipocyte-cancer
 cells or specifically in the isolated adipocytes from the co-culture system?

Our co-culture system is performed on a trans-well culture plate, which consists of a
 permeable membrane separating two cell populations. This allows the exchange of soluble
 factors while isolating each population in distinct layers. As a result, our q-PCR and WB
 analyses could be exclusively performed on each cell population. For clarity, we have
 revised the method section of the manuscript (line 430-432).

3. The provided Figure Legends are quite concise, making it hard to determine how the
 experiments were exactly performed.

The Figure Legend section has been updated with additional details.

**Reviewer #2 (Remarks to the Author):**

In this manuscript, the authors employed an adipose tissue-specific BECN1 knockout
 (BaKO) mouse model to investigate the effects of dysfunctional adipocytes (caused by loss-
 of-function of Beclin1) on the growth of certain types of tumors. They reported that BaKO
 adipocytes promote tumor growth both in vitro and in vivo using a combination of multiple
 mouse models. Furthermore, they demonstrated the involvement of the YAP/TAZ signaling
 pathway in tumor progression. As a demonstration, deletion or inhibition of YAP/TAZ in the
 BaKO adipocytes restored normal cell functions, suppressed lipodystrophy, and mitigated
 inflammatory phenotypes. Overall, the data are robust, and the findings are intriguing. The
 conclusions are well-supported by their results. However, there are significant concerns
 regarding unclear presentations and a lack of sufficient mechanistic studies, which
 somewhat diminish the enthusiasm for this otherwise interesting study. The following
 concerns, including both major and minor issues, are outlined below:

We appreciate the positive feedback and interest expressed by the reviewer. Acknowledging
the validity of the reviewer's concerns, we have made deliberate attempts to address them
within the manuscript. Specifically, the molecular insights were clarified to resolve any
obscurity in our text. We believe these efforts have enriched our discussion section and the
overall contents of the manuscript.

Major Concerns:

1). Several result titles start with "Dysfunctional adipocytes," which fails to provide precise
information about the adipocytes in the respective paragraph. If these cells are BECN1 KO
cells, it is recommended to use "BaKO adipocytes."; If the cells were isolated from diet-
induced obese adipose tissue, then the term "diet-induced obese adipocytes" should be
used. Employing clearer terminology would help readers better understand the nature of
these cells.

We appreciate the reviewer's observation. Upon reviewing our manuscript, we acknowledge
that the use of "Dysfunctional adipocytes" in several result titles may lead to ambiguity. To
address this concern, we have revised the titles to provide more specific information.

2). It has been reported that loss-of-function of BECN1 promotes tumor growth in cancer
cells. Similarly, here, the authors found that BaKO in adipocytes stimulates co-cultured
tumor cells and activates downstream YAP/TAZ pathways, paralleling the role observed
previously in cancer cells. The stimulation of the YAP/TAZ signaling pathway might explain
the impact on adipogenesis reduction and inflammation enhancement. However, one
significant effect of BaKO in adipocytes is enhanced lipolysis (Fig. 5g, Supplementary Fig.
5e and i). Considering that lipolysis can promote cancer growth, it's surprising that the
authors didn't investigate the mechanism underlying how BaKO adipocytes increase
lipolysis.

We acknowledge that it is crucial to understand the mechanisms underlying how BaKO
adipocytes increase lipolysis, considering its potential role in promoting cancer growth. Our
investigation revealed that BECN1-deficient adipocytes exhibit enhanced lipolysis, as
evidenced by the phosphorylation of multiple sites on hormone-sensitive lipase (HSL).
Notably, HSL phosphorylation is commonly mediated by PKA (at S563, S660) and AMPK (at
S565)[15], indicating heightened PKA activity and suppressed AMPK in BECN1-depleted
adipocytes.

The issue is to determine how much lipolysis contributes to tumor progression in our models.
To investigate the impact of lipolysis in-vivo, we conducted RNA-seq analysis on tumor cells
transplanted in WT, BaKO, and BYTaKO. If the tumor cells rigorously utilized the lipid
contents secreted from active lipolysis, their genes associated with lipid metabolism would
have increased. Heatmap analysis focusing on genes associated with fatty acid (FA)
metabolism revealed that tumors grown in BaKO displayed heightened FA metabolism,
confirming increased lipolysis and FA release (**Revised Fig. 8a**). However, this
enhancement was not restored in BYTaKO tumors, suggesting that factors other than
lipolysis contribute significantly to the observed tumor regression in BYTaKO.

a

GSEA (Hallmark): FA METABOLISM

**Revised Fig. 8a** – Heatmap analysis of genes associated with FA metabolism. Bulk RNA-
seq was conducted on MC38 tumors grown in WT, BaKO, and BYTaKO mice for 2 weeks.

While we acknowledge that lipolysis may have played a role in tumor progression in BaKO,
our data suggest it is not the sole major factor. Importantly, we identified Lipocalin 2 (LCN2)
as a key molecule secreted from BECN1-deficient adipocytes, promoting tumor progression.
Deletion of the LCN2 receptor in cancer cells prevented the rapid growth induced by
BECN1-deficient adipocyte co-cultivation (Main Fig. 2j). This underscores the complexity of
the interactions, where lipolysis, though contributory, is not the primary driver of tumor
growth in our models.

3). The manuscript lacks crucial information about the recipient mice for co-injected WT or
*Becn1* KO cells with 4T1. Are these nude mice with compromised inflammatory reactions for
the injected cancer cells? Additionally, the nature of the co-injected fat cells remains unclear,
as there is mention of imSVCs (line 99) and differentiated adipocytes (line 100).

For the adipocyte-cancer co-injection system, we utilized immunodeficient mice (N2G). This
information has been incorporated into both the Results and Methods sections of the
manuscript. To provide further clarity, we have expanded on the schematic details of the co-
injection system in the main text (result section, line 114-118; Method section, line 411-413).
Regarding imSVC establishment, please refer to our previous research article [16].

line 114-118

we established immortalized stromal vascular cell (imSVC) lines with conditional *Becn1* KO system
(ROSA-Cre^{ERT2}). imSVCs were fully differentiated into adipocytes and then treated with 4-
hydroxytamoxifen (4-OHT) to deplete BECN1. Next, we co-injected WT or *Becn1* KO imSVCs and
breast cancer cells (4T1) into the mouse flanks

line 411-413

Immunocompromised N2G (NOD; *Prkdc*^{em1Gmcr}; *Il2rg*^{em1Gmcr}) mice were obtained from Gemcro Corp
(Seoul, Republic of Korea). These mice were generated from NOD with deletion of entire exons of
PRKDC and IL2RG.

4). There's a need to reconcile the observations of BECN1-deficient adipocytes and HFD-
feeding induced obese adipocytes. In Fig 6, the authors note that inhibition of YAP/TAZ
activity in both BECN1-deficient and HFD-fed obese adipocytes yields the same tumor
suppression effect. However, these two adipocyte types exhibit stark morphological
differences: BECN1-deficient adipocytes have small lipid droplets and are prone to
dedifferentiation to fibroblasts, while HFD-induced obese adipocytes possess unusually
large lipid contents and do not transform into fibroblasts. Moreover, there's confusion in line

298-299, where they state that "the WATs of the HFD-fed mice had decreased levels of
 BECN1," while these fat cells do not share the lipid droplet phenotype of BECN1 KO
 adipocytes. Addressing these discrepancies is crucial.

Thank you for highlighting this critical aspect. We acknowledge the noticeable morphological
 differences between BECN1-deficient adipocytes and HFD-induced obese adipocytes.
 Despite these distinctions, we were able to identify certain shared phenotypes between the
 two samples. BECN1-deficient adipocytes exhibit reduced expression of adipogenic genes
 and elevated levels of inflammatory signals (Main Fig. 2a-c, 5f). These features are known to
 be analogous to the HFD-induced obese adipocytes [17, 18]. To further characterize these
 adipocytes, we conducted RNA-seq from HFD-induced obese adipose tissue. Next, we
 compared the HGSEA results between HFD-fed and BaKO WATs. The significantly
 upregulated / downregulated gene sets were arranged regarding their normalized
 enrichment score (NES), and gene sets that commonly appear in BaKO WAT were marked
 with yellow stars (**Revised Fig. 9a**). Both BaKO WAT and HFD WAT shared inflammatory
 phenotypes followed by downregulation of genes associated with adipogenesis, FA
 metabolism, and cholesterol homeostasis [18]. Additionally, having EMT signaling as the
 second most significantly upregulated gene set aligns well with what we observed not only in
 BaKO WAT but also in Peritumoral WAT (Supplementary Fig. 3b). Presumably, HFD-fed
 adipocytes acquire partially dedifferentiated phenotypes, as observed in BECN1-deficient
 adipocytes and CAAs. Thus, though BECN1-deficient adipocytes and HFD-induced obese
 adipocytes evidently differ in their morphologies, they share some common features in terms
 of adipocyte quality.

**Revised Fig. 9a** – HGSEA result of RNA-seq conducted on mice fed with HFD (60% fat
 478 kcal) for 12 weeks. Significantly altered gene sets were arranged according to NES. Yellow
 stars indicate the commonly altered gene sets in the HGSEA result of BaKO WAT.

The effectiveness of the YAP/TAZ inhibitor (verteporfin) on both BaKO and HFD-fed mice is
 explained by the high expression of YAP/TAZ in their adipocytes. We have shown that
 BECN1 depletion in adipocytes induce YAP/TAZ expression (Main Fig. 5). This phenomenon
 is crucial when determining the differentiation state, leading to disrupted adipocyte
 homeostasis and causing formation of malignant TME to promote tumor (Main Fig. 1) [19,
 20]. Similarly, WATs of HFD-fed mice contained high levels of YAP/TAZ (Main Fig. 7e) [21].
 Hence, we hypothesized that YAP/TAZ^{high} adipocytes in the TME contribute to tumor

progression and that verteporfin treatment could induce tumor regression both in BaKO and
HFD-fed mice.

We added the following paragraph to address the discrepancy about the morphological
difference between BECN1-deficient adipocytes and HFD-induced obese adipocytes in our
discussion section (line 333-338).

line 333-338

Of note, BaKO WAT displays lipodystrophy phenotypes, such as shrinking adipocytes and lipid
contents, which significantly differ from obese WAT. However, both conditions were suitable for rapid
tumor progression and were sensitive to VP treatment. We identified multiple features of BaKO WAT
that are analogous to those of obese WAT (Supplementary Fig. 9e). This highlights that instead of fat
mass, the quality of adipose tissue, pertaining to inflammation, vascularity, adipokine secretion, and
extracellular matrix integrity, is more crucial when evaluating the TME status.

5). The autophagy-independent effect of BECN1 in adipose tissue needs further
experimentation to provide solid support. While the authors discuss the involvement of key
factors such as LATS1 and MST1 (lines 303 to 312), more experiments are required to
establish their role in regulation.

We appreciate the reviewer's insightful comment and acknowledge the need for additional
experimentation to solidify the autophagy-independent effect of BECN1 in adipose tissue. In
response to Reviewer #1's queries regarding the dispensability of autophagy for BECN1-
YAP/TAZ regulation, we conducted a series of experiments to provide more robust evidence.
Specifically, we demonstrated that inhibiting autophagy alone (using HCQ treatment) does
not suffice to enhance YAP/TAZ levels, resulting in accelerated tumor growth (**Revised Fig.**
**4a-d**).

As the reviewer mentioned, we have only roughly discussed the relationship between
BECN1 and other Hippo-associated proteins such as LATS1 and MST1. To further explore
the interaction between BECN1 and other Hippo-associated proteins, we conducted PLA
which indicates the protein-protein interactions determined by their proximity [4].
Consistently, we found that BECN1 makes moderate interactions with LATS1 and MST1
(**Revised Fig. 3d, e**). However, when we checked the interaction between BECN1 and
MOB1, its intensity exceeded all other interactions between BECN1 and Hippo-associated
proteins (**Revised Fig. 3b**).

Next, we investigated the physiological outcome of BECN1-MOB1 interaction in adipocytes.
As we depleted BECN1 from adipocytes, there was suppression of Hippo pathway
demonstrated by diminished interactions between MOB1-LATS1 and LATS1-YAP1
(**Revised Fig. 3g, h**). These results prompted us to hypothesize whether BECN1 regulates
MOB1 phosphorylation. However, BECN1 rather altered total MOB1 level instead of its
phosphorylation rate (**Revised Fig. 3i**), and this phenomenon was also observed in BaKO
mouse iWAT (**Revised Fig. 3j**). Finally, overexpression of *BECN1*, *YAP1*, and *MOB1* in
HEK293 cells revealed that BECN1 regulates YAP phosphorylation through MOB1 (**Revised**
**Fig. 3k**). Taken together, we could uncover the mechanistic insight into how BECN1
contributes to Hippo pathway. Accordingly, we revised our summary figure of the manuscript
(**Revised Fig. 3l**). We hope these comprehensive findings address the reviewer's concerns
and enhance the clarity of our study.

The results have been incorporated into our Main Figure 5 and Supplementary Figure 9.

6). The discussion lacks depth and detail. The authors should provide more comprehensive
mechanistic insights into the effects of BECN1 in adipocytes. Particularly, they should

elucidate how BECN1 suppresses the YAP/TAZ signaling pathway specifically in adipocytes
and compare their findings about BECN1/YAP/TAZ signaling in adipocytes with previous
reports in other cell types.

We appreciate the reviewer's concern and admit that our discussion section needs more
improvement. Recognizing this, we have expanded the discussion section to encompass the
following key points: (1) Exploring the contribution of CAAs to cancer cachexia and adipose
tissue wasting, (2) Contrasting the morphologies of BaKO and HFD adipocytes, noting
shared features and their relevance to verteporfin response, (3) Unveiling mechanistic
insights into the BECN1-MOB1-YAP/TAZ-LCN2 axis in adipocytes, and (4) Investigating the
potential impact of BECN1 loss in other stromal cells, such as fibroblasts in the tumor
microenvironment (TME), and its role in promoting tumor progression (line 321-396)

Interestingly, the shared interest among all three reviewers in the potential impact of the
BECN1-YAP/TAZ axis in various cell types prompted us to delve deeper into the effect of
BECN1-deficient fibroblasts and their influence on cancer growth. Our findings indicate that
BECN1-deficient fibroblasts can enhance cancer proliferation rates and induce epithelial-
mesenchymal transition (EMT) (**Revised Fig. 6b-h**). While we cannot definitively assert that
the BECN-YAP/TAZ axis is the sole modulator for these effects on cancer cells, we
emphasize the critical role of BECN1 levels in fibroblasts in shaping a malignant TME.

**The results have been incorporated into Supplementary Figure 11.**

Minor Concerns:

1). The RNA-seq results (Fig. 2a&b) were obtained from iWAT, which includes multiple cell
populations. However, BECN1 deficiency is specific to adipocytes in the isolated fat tissue.

The reviewer's concern is valid, noting that iWAT consists of multiple cell populations,
including preadipocytes, immune cells, fibroblasts, and endothelial cells [22]. Therefore,
performing bulk RNA-seq could lead to misleading interpretations. To address this, we
verified the RNA-seq results in-vitro. Indeed, Main Fig. 2b was performed in-vitro to confirm
that the significantly upregulated gene sets, TNF α signaling and inflammatory response,
were due to the alterations in adipocytes (not due to the interference of immune cells).
Similarly, the regulations of YAP downstream signaling and adipogenic genes (Main Fig. 3h-
j) were verified in-vitro (Main Fig. 5f-i).

2). In line 165, clarify whether "Co-regulated" means "Co-upregulated" or "Co-
downregulated."

This has been fixed in the main text. (line184-186)

**In concordance, the genes associated with adipogenesis and fatty acid (FA) metabolism were**
**consistently downregulated, supporting the concept of mesenchymal transition of adipocytes**

3). In line 219, clarify what "restored phenotypes" refer to. Also, given that the KO effects
are specific to adipocytes, it's important to clarify that the liver phenotypes aren't a direct
result of YAP/TAZ KO.

This has been fixed in the main text. (line 258-261)

**The results showed impaired autophagy (Supplementary Fig. 7b); however, lipodystrophy phenotypes**
**observed in BaKO were restored in BYTaKO (Fig. 6b-g). This process involved the restoration of**
**adipose tissue mass, potentially leading to the alleviation of liver steatosis through successful lipid**
**storage in the adipose tissue**

4). Line 279: Clarify the specific KO mouse models used in the study.

This has been fixed in the main text. (line 303)

To complement our genetic mouse model (BYTaKO), we employed VP to inhibit YAP/TAZ activity in
adipose tissue

5). In line 290, specify which phenotypes are meant in relation to the effects of BECN1
depletion.

This has been fixed in the main text. (line 324-327)

While only Wnt and Notch signaling have been identified as the mediators of adipocyte
dedifferentiation for CAA development, we show that similar phenotypes, such as suppression of
adipogenic markers and acquisition of fibroblast-like features, are obtained from BECN1 depletion

6). In line 298, explain what is meant by the "quality of adipose tissue." Does "quality" refer
to the capacity for lipolysis, appropriate adipokine secretion, or another aspect?

We also acknowledge the usage of word 'quality' is very vague in the context. However,
instead of removing the term entirely, we have provided additional details of the crucial
features that may contribute to adipose tissue homeostasis. This way, we can suggest
important 'qualities' of adipose tissue to consider when evaluating TME.

This has been fixed in the main text. (line 336-338)

This highlights that instead of fat mass, the quality of adipose tissue, pertaining to inflammation,
vascularity, adipokine secretion, and extracellular matrix integrity, is more crucial when evaluating the
TME status

**Reviewer #3 (Remarks to the Author):**

Summary:

This paper demonstrates Becn1 conditional KO in adipocytes can promote tumor growth of
breast cancer and colorectal cancer cells in vivo and in vitro. Adipokine array analysis of
serum plasma from Becn1 mutant mice, and RNA sequencing of white adipose tissue of
mutant and control mice revealed upregulation of LCN2 and TNF α signaling in mutants in
concordance with the tumor growth phenotype. Further analysis suggests that BECN1 KO in
adipocytes creates a pro-tumorigenic environment by eliciting gene transcription and
markers associated with CAA (cancer associated adipocyte) status. Upregulation of
YAP/TAZ gene signatures was observed in both BECN1 KO and peritumoral adipocytes,
suggesting a potential mechanism for the tumor growth phenotypes observed. Beclin KO
mice also lacking YAP and TAZ reversed many lipodystrophy phenotypes observed, as well
as the increase in TNF α levels. Since Atg7 conditional KO did not elicit similar YAP/TAZ or
tumor growth phenotypes in vitro or in vivo that were observed in BECN1 KO, the authors
speculate that perhaps BECN1 function via PI3K/MTOR may be responsible for some of the
phenotypes. It is my opinion that this work is novel and important, and suitable for
publication in this journal with some revisions to characterize the phenotypes and bolster
molecular data.

We appreciate the reviewer's thorough understanding of our manuscript and thank them for
their positive perspective and constructive critiques. The acknowledgment of the novelty and
importance of our work is encouraging. We are committed to addressing the suggested
revisions to further characterize the phenotypes and enhance the molecular data. Your
feedback is invaluable, and we will diligently incorporate the necessary improvements to
ensure the manuscript meets the standards of this journal.

Major Comments:

1. It would be good to look at LCN2 (TNF α ?) in the context of ATG7/ Baf/HCQ in vitro.

To enhance clarity, we have incorporated TNF α downstream and LCN2 expression data
following Baf/HCQ treatment (**Revised Fig. 10a**). Similar to YAP downstream result
(Supplementary Fig. 4c), we could not observe a significant difference in TNF α downstream
in adipocytes treated with Bafilomycin or HCQ. Although there was a moderate increase in
LCN2 expression, it was not as extreme as observed in BECN1-deficient adipocytes (Main
Fig. 2b and g).

**Revised Fig. 10a** – Relative qPCR result of TNF α downstream and LCN2 in adipocytes
treated with HCQ or Bafilomycin. Autophagy inhibitors were treated (48 hr) once adipocytes
were fully differentiated.

The result has been incorporated into our Supplementary Figure 4.

2. Fig 1h, add p62 and/or I ϵ 3b to western blot, to help add context for the Atg7 data.

We thank the reviewer for this comment, as BECN1-mediated autophagy has not been
extensively addressed in the manuscript. Thus, we provide the data that autophagy was
blocked in our imSVC cell line (**Revised Fig. 11a**).

**Revised Fig. 11a** – Ablation of BECN1 leads to autophagy inhibition in differentiated
 imSVCs. Fully differentiated imSVCs were treated with 4-OHT to deplete BECN1.

**The result has been incorporated into our Main Figure 1.**

3. Along with changes in tumor cell proliferation, it would be good to evaluate changes in
 cell death, i.e. by CC3 staining (i.e. Fig 6), since autophagy can regulate cell death.

We apologize for any confusion regarding the experimental design we implemented in Fig.6.
 As the reviewer mentioned, autophagy ablation does play a crucial role to regulate cell
 survivability [23]. However, our manuscript mainly covers BECN1 depletion in adipocytes not
 in cancer cells [24]. It's worth noting that we have assessed adipocyte death in our previous
 report. We have shown that BECN1-deficient adipocytes experienced persistent
 accumulation of endoplasmic reticulum (ER) stress [24]. Furthermore, we have used TUDCA
 to relieve ER stress-mediated adipocyte death [24].

Nevertheless, we assume the reviewer was curious how cell death signal is altered in cancer
 cells when affected by adipocytes. It is reasonable to examine alterations in apoptosis, as
 Main Fig. 6j illustrates that BYT-deficient adipocytes could inflict apoptosis in cancer cells.
 Therefore, we conducted RNA-seq on the tumor cells transplanted in WT, BaKO, and
 BYTaKO. Here, we provide heatmap analysis for the expression pattern of genes associated
 with apoptosis. Surprisingly, tumors grown in BaKO exhibited downregulation of apoptosis;
 however, this effect was not significantly restored in cancer cells grown in BYTaKO
 **(Revised Fig. 12a)**. Taken together, apoptosis induction is not considered the major factor
 to modulate tumor regression in BYTaKO. Instead, factors relating to cell proliferation such
 as adipokine secretion or environmental cues could drive the outcome. We hope these
 clarifications provide a better understanding of our experimental approach and results.

**Revised Fig. 12a** - Heatmap analysis of genes associated with apoptosis. Bulk RNA-seq
 was conducted on MC38 tumors grown in WT, BaKO, and BYTaKO mice for 2 weeks.

4. Please comment/discuss on YAP/TAZ inhibitor effect on other stromal cells? Could be
relevant given this paper: The β -catenin/YAP signaling axis is a key regulator of melanoma-
associated fibroblasts

First, we would like to express gratitude to the reviewer about the constructive comment. We
admit that BECN1-YAP/TAZ axis is of great importance, especially in other stromal cells that
are populated in TME. Before getting into this matter, we would like to share the mechanistic
insight that was revealed throughout this study.

The Hippo pathway involves a cascade of kinase activities of different molecules such as
MST1, LATS, and MOB1 [25]. Here, YAP/TAZ are tightly controlled by their phosphorylation
status modulated by Hippo-associated proteins. Therefore, alteration of cellular YAP/TAZ
level frequently involves alterations in kinase activities of Hippo-associated proteins. To
identify the direct impact of BECN1 on Hippo-associated proteins, we conducted PLA which
indicates the proximity between two proteins of interest [4]. PLA result between BECN1 and
Hippo-associated proteins has shown modest interactions between BECN1-LATS1 and
BECN1-MST1 interactions (**Revised Fig.3d, e**). However, when we checked the interaction
between BECN1 and MOB1, its intensity exceeded all other interactions between BECN1
and Hippo-associated proteins (**Revised Fig. 3b**). Consistently, BECN1-deficient adipocytes
exhibited downregulation of MOB1, indicating that loss of BECN1 leads to suppression of
Hippo signaling through MOB1 (**Revised Fig. 3i**). This was confirmed in BaKO mouse iWAT
(**Revised Fig. 3j**).

Implication of BECN1-MOB1-YAP/TAZ axis was also tested in preadipocytes, which are
expected to have fibroblast-like features. BECN1-deficient preadipocytes also exhibited
suppression of Hippo signaling demonstrated by the reduction in MOB1-LATS1 interaction
(**Revised Fig. 6a**). Furthermore, we have shown that BECN1-deficient fibroblasts are
sufficient to induce cancer malignancy through enhanced proliferation and EMT (**Revised**
**Fig. 6b-h**). Therefore, usage of YAP/TAZ inhibitor could be effective on both adipocytes and
fibroblasts in TME, which suggests somewhat broadened effects of verteporfin on the TME
shown in Main Fig. 7d-g.

As the reviewer provided the paper, Wnt/ β -catenin activity can possibly modulate YAP/TAZ
in adipocytes [25, 26]. Our Main Fig. 5a-e also supports that there is a similar expression
pattern between β -catenin and YAP/TAZ. We believe this discussion is highly relevant to our
study and should be continued in the next reviewer's comment (Q5). Once again, we thank
the reviewer for providing a thoughtful insight with a relevant literature.

5. Are there any changes in Wnt signaling given the beta catenin changes? Can you
comment on IL6 given its previous role? Do you think IL6 changes could also drive
phenotypes?

As previously mentioned, Wnt/ β -catenin signaling can be a strong candidate to control
YAP/TAZ [25, 26]. Here, we provide heatmap analysis of Wnt/ β -catenin signaling in WT,
BaKO, and BYTaKO iWATs. The result was very striking, as it was highly activated in BaKO
while the restoration in BYTaKO was evident (**Revised Fig. 13a**). Such trend implies that
Wnt/ β -catenin signaling can be regulated by YAP/TAZ activity. In fact, there were multiple
reports demonstrating the crosstalk between Hippo and Wnt signaling. Certainly, a few
investigated how Hippo signaling triggered Wnt/ β -catenin pathway [27-30].

**Revised Fig. 13a** – Heatmap analysis of genes associated with Wnt/ β -catenin signaling.
 Bulk RNA-seq was conducted on WT, BaKO BYTaKO iWATs.

IL-6 has been reported to induce Wnt/ β -catenin activity through JAK2 and FOXO3 in
 colorectal cancer (CRC) and kidney, respectively [31, 32]. Our RNA-seq result on BaKO
 WAT also illustrated that IL6-JAK-STAT3 signaling is the 5th most significantly upregulated
 gene set (Main Fig. 2a). Additionally, IL-6 itself was upregulated in BaKO WAT while
 restored in BYTaKO WAT (Main Fig. 6f). To resolve whether YAP/TAZ activity is capable of
 regulating IL6 expression, we checked the CHIP-seq result on IL6 loci. Interestingly, YAP-
 TEAD are found to be bound at the promoter region of IL6. Therefore, we can conclude that
 YAP-TEAD-IL6-JAK-Wnt/ β -catenin axis is a plausible underlying mechanism in our study.
 Subsequent investigation should be performed to validate this pathway, and we have
 incorporated this aspect into the text; the following paragraph has been inserted in line 358-
 371 in the discussion section.

**Revised Fig. 13b** - ChIP-seq result of YAP and TEAD in MCF10A and PC-9 cells at the
 genomic loci of IL6. ChIP-seq results were collected from 'chip-atlas.org'. Intensified peaks
 indicate potential binding region of the indicated proteins at those chromatin regions.

line 358-371

Our transcriptomic analyses on BaKO and peritumoral WATs well-characterized the dysfunctional
 adipocytes and identified IL-6-JAK-STAT3 signaling as the 5th and 9th most significantly upregulated
 gene sets (Fig. 2a, 3f) in BaKO and peritumoral WATs, respectively. Previous studies have shown that,
 IL-6 expression is not only relevant to the pro-inflammatory signals released from impaired adipose
 tissue, but also relevant to the induction of Wnt/ β -catenin activity through JAK2 and FOXO3. In this
 study, we showed the upregulation of IL-6 in BaKO WAT while it was restored in BYTaKO WAT (Fig. 6f).
 Additionally, the CHIP-seq results of YAP and TEAD indicate potential binding regions at the IL-6
 promoter, suggesting that the activation of YAP directly regulates IL-6 expression (Supplementary Fig.

10a). Heatmap analysis on Wnt/ β -catenin signaling in WT, BaKO, and BYTaKO WATs revealed a
significant activation in BaKO but restoration in BYTaKO (Supplementary Fig. 10b). This observed trend
aligns well with our study, wherein lipodystrophy phenotypes and tumor progression were restored in
BYTaKO (Fig. 6, 7). These findings suggest the IL6-JAK-Wnt/ β -catenin axis as a plausible underlying
mechanism for adipocyte transformation; however, further investigations are needed to validate this
finding.

6. Is LCN2 or the receptor expressed in the tumor cells of interest? Consider providing
some staining and/or citations. (same for TNFa/ receptor?) Is there any other previous data
regarding LCN2 in breast/colorectal cancer specifically?

We found that the available literature on LCN2R (SLC22A17) is not sufficient to provide
information on its expression across various mouse cancer cell lines. However, we could
obtain LCN2R expression data for human cancer cell lines from the Cancer Cell Line
Encyclopedia (<https://sites.broadinstitute.org/ccl/>). The expression of LCN2R varied across
different tissues, with fibroblasts notably displaying relatively high levels of LCN2R. Human
breast and bowel-related cancer cell lines exhibited a wide range of LCN2R expression
distribution (**Revised Fig. 14a**).

**Revised Fig. 14a** – Distribution of LCN2R expression across multiple human cancer cell
lines with various tissue origin.

LCN2 is known to promote breast and colon cancer progressions in both human and mouse
[33-35]. When detecting the basal LCN2R expression across multiple cell lines, we observed
considerable variation in its expression (**Revised Fig. 14b**). Notably, it was not necessarily
upregulated in breast and colon cancers. Our thoughts on this phenomenon will be further
discussed in Q7 and Q9.

**Revised Fig. 14b** – qPCR result of LCN2R on multiple cancer cell lines that were used in
 our in-vivo study.

7. Please mention whether obesity is predictive of poor outcome in breast and colon cancer
 respectively. Overall, the discussion of cancer is very general, some more specifics for
 breast and colon (as well as cancers where there was no difference!) would be useful.

To date, obesity has been related to 13 different kinds of cancers, including breast and colon
 cancers [36]. Additionally, recent epidemiology on BMI and cancer risk identified that longer
 duration, greater degree, and younger age of onset are obese-related factors that correlate
 with cancer incidence [37]. Especially, longitudinal exposure to high BMI associated with 12
 cancers: corpus uteri, kidney, gallbladder, myeloma, leukemia, breast, colorectal, liver,
 thyroid, brain, and head/neck cancers [36]. Of these cancers, tissue origins such as the
 kidney, bone marrow, breast, colorectal, and liver have a substantial presence of adipocytes
 within their environment. This suggests that some cancers, due to their proximity to large
 population of adipocytes, ‘adipose health’ becomes pivotal in determining the TME
 malignancy. Hence, we believe that the CAA literature will provide a profound understanding
 to obesity-related cancers.

We thank the reviewer for pointing out the obscurity regarding cancer specificity in our
 manuscript. We have noted that both breast and colon cancers are surrounded by
 substantial amount of adipose tissue, thereby being influenced by adipose milieu as a
 component of the TME. Considering the increased likelihood of interaction between cancer
 cells and adipocytes, we hypothesized a heightened crosstalk between these two cell types.
 This interaction is expected to lead to a more robust transformation of adipocytes, potentially
 increasing their tumorigenicity. Given our emphasis on LCN2 as a key factor derived from
 adipocytes that promotes breast and colon cancers, we assessed LCN2 levels in adipocytes
 cocultured with cancer cells. Notably, cocultured adipocytes exhibited elevated levels of
 LCN2 (**Revised Fig. 15a**). This finding supports the notion that with a greater presence of
 adipocytes in the TME, these cancers may become more susceptible to metabolites and
 cytokines derived from adipocytes.

**Revised Fig. 15a** – Immunocytochemistry staining LCN2 level in preadipocytes, mature
 adipocytes, and cancer-co-cultured adipocytes. Adipocytes were co-cultured with EO771
 cells for 2 days.

**The results have been incorporated into our Supplementary Figure 3.**

8. EMT: all the data for this are RNA levels/ gene signatures. Please add some protein data
 as well for a few markers. Please discuss, are EMT signatures associated with CAA status?

We admit that EMT markers should be confirmed through additional techniques.

Accordingly, we stained Vimentin and E-cadherin in BECN1-deficient adipocytes. In line with
 RNA-seq results, BECN1-deficient adipocytes exhibit mesenchymal transition (**Revised Fig.**
 **16a**).

**Revised Fig. 16a** – Immunocytochemistry staining E-Cadherin (red) and Vimentin (green) in
 differentiated adipocytes treated with or without doxycycline.

CAAs are widely recognized for their propensity to undergo dedifferentiation, involving
 morphological alterations resulting in fibroblast-like and mesenchymal features [38]. Yiwei Li
 et al. referred to this transformation of adipocytes as adipocyte derived fibroblasts (ADF)
 [10]. In another recent study on CAAs, the term ‘adipocyte mesenchymal transition (AMT)’

was introduced [39]. Consequently, the acquisition of a mesenchymal signature well-
characterizes CAA status.

9. Any thoughts as to why the 2 other tumor cell types did not show the same phenotypes?

Building on our response on Q6 and Q7, the basal LCN2R expression between different
cancer cell lines (**Revised Fig. 14b**) does not explain their sensitivity to BECN1-deficient
adipocytes in-vivo (Main Fig. 1). Therefore, we hypothesized that LCN2R expression could
be altered situationally. To investigate the LCN2R level in our cell line of interest (EO771),
we detected change in its expression upon adipocyte co-culture. While EO771 co-cultured
with WT adipocytes show modest amount of LCN2R, EO771 co-cultured with BECN1-
deficient adipocytes exhibited remarkably amplified levels of LCN2R (**Revised Fig. 17a**).
This result suggests that LCN2R expression could be responsive to LCN2 accessibility.
Additionally, whether other cancer cell types are capable of modifying adipocyte-LCN2, as
observed in Revised Fig. 15a, remains unanswered. Therefore, the reason why the 2 other
tumor cell types were not responsive to BECN1-deficient adipocytes could be attributed to
the dynamic crosstalk between adipocytes and cancer cells, regulating LCN2 and LCN2R.

**Revised Fig. 17a** – LCN2R expression in EO771 co-cultured with WT or BECN1-deficient
adipocytes. Cancer cells were co-cultured with adipocytes for 2 days.

This figure has been added into our Supplementary Figure 2

Minor comments:

1. Spell out LCN2 fully the first time it appears in the text.

This has been fixed in the main text.

2. Line 151 use “co-culture” instead of co cultivation

This has been fixed in the main text.

3. Should mention/discuss and cite Beige Adipocyte Maintenance Is Regulated by
Autophagy-Induced Mitochondrial Clearance paper

This content has been added in line 200-202.

4. Please spell out what are imSVCs the first time it's mentioned since I'm not familiar with
this acronym.

This has been fixed in the main text. imSVC establishment procedure is further elaborated in
the method section.

5. Supp fig 4a: please add quantification

Due to redundancy (with Main Fig. 4a), Supplementary Fig. 4a has been removed.

6. Supp. Fig 5c could benefit from BODIPY stain for ease of interpretation.

We appreciate the reviewer for positing a critical point. However, we would like to highlight
our earlier publication has already demonstrated the development of a liver steatosis
phenotype in BaKO attributed to lipodystrophy. Consequently, Supp. Fig 5c adequately
illustrates the lipid accumulation in liver. Reference was added in line 111.

7. Fig. 6g: is WT versus Beclin/YT CKO significantly different? Please add this to the figure.

This has been fixed in the main figure.

- 1. Corley, S.M., et al., *Plau and Tgfb3 are YAP-regulated genes that promote keratinocyte*
*proliferation*. Cell Death Dis, 2018. **9**(11): p. 1106.
- 2. Kurppa, K.J., et al., *Treatment-Induced Tumor Dormancy through YAP-Mediated*
*Transcriptional Reprogramming of the Apoptotic Pathway*. Cancer Cell, 2020. **37**(1): p. 104-+.
- 3. He, L.Z., et al., *YAP and TAZ are transcriptional co-activators of AP-1 proteins and STAT3*
*during breast cellular transformation*. Elife, 2021. **10**.
- 4. Fredriksson, S., et al., *Protein detection using proximity-dependent DNA ligation assays*. Nature
Biotechnology, 2002. **20**(5): p. 473-477.
- 5. Ni, L.S., et al., *Structural basis for Mob1-dependent activation of the core Mst-Lats kinase*
*cascade in Hippo signaling*. Genes & Development, 2015. **29**(13): p. 1416-1431.
- 6. Marquez, R.T. and L. Xu, *Bcl-2:Beclin 1 complex: multiple, mechanisms regulating*
*autophagy/apoptosis toggle switch*. American Journal of Cancer Research, 2012. **2**(2): p. 214-
221.
- 7. Fernández, A.F., et al., *Disruption of the beclin 1-BCL2 autophagy regulatory complex promotes*
*longevity in mice (vol 558, pg 136, 2018)*. Nature, 2018. **561**(7723): p. E30-E30.
- 8. Song, T.X. and S.H. Kuang, *Adipocyte dedifferentiation in health and diseases*. Clinical Science,
2019. **133**(20): p. 2107-2119.
- 9. Zoico, E., et al., *Adipocytes WNT5a mediated dedifferentiation: a possible target in pancreatic*
*cancer microenvironment*. Oncotarget, 2016. **7**(15): p. 20223-20235.
- 10. Li, Y.W., et al., *Compression-induced dedifferentiation of adipocytes promotes tumor*
*progression*. Science Advances, 2020. **6**(4).
- 11. Bi, P.P., et al., *Notch activation drives adipocyte dedifferentiation and tumorigenic*
*transformation in mice*. Journal of Experimental Medicine, 2016. **213**(10): p. 2019-2037.
- 12. Sahai, E., et al., *A framework for advancing our understanding of cancer-associated fibroblasts*.
Nature Reviews Cancer, 2020. **20**(3): p. 174-186.
- 13. Wu, P.H., et al., *Single-cell morphology encodes metastatic potential*. Science Advances, 2020.
**6**(4).
- 14. Cadart, C., et al., *Exploring the Function of Cell Shape and Size during Mitosis*. Developmental
Cell, 2014. **29**(2): p. 159-169.
- 15. Djouder, N., et al., *PKA phosphorylates and inactivates AMPK α to promote efficient lipolysis*.
Embo Journal, 2010. **29**(2): p. 469-481.
- 16. Jin, Y., et al., *Depletion of Adipocyte Becln1 Leads to Lipodystrophy and Metabolic*
*Dysregulation*. Diabetes, 2021. **70**(1): p. 182-195.
- 17. Nadler, S.T., et al., *The expression of adipogenic genes is decreased in obesity and diabetes*
*mellitus*. Proceedings of the National Academy of Sciences of the United States of America,
2000. **97**(21): p. 11371-11376.
- 18. Jiang, N., et al., *Cytokines and inflammation in adipogenesis: an updated review*. Frontiers of
Medicine, 2019. **13**(3): p. 314-329.
- 19. Yu, F.X., et al., *Protein kinase A activates the Hippo pathway to modulate cell proliferation and*
*differentiation*. Genes & Development, 2013. **27**(11): p. 1223-1232.
- 20. Hong, J.H., et al., *TAZ, a transcriptional modulator of mesenchymal stem cell differentiation*.
Science, 2005. **309**(5737): p. 1074-1078.
- 21. Shen, H.Y., et al., *The Hippo pathway links adipocyte plasticity to adipose tissue fibrosis*. Nature
Communications, 2022. **13**(1).
- 22. Emont, M.P., et al., *A single-cell atlas of human and mouse white adipose tissue (vol 603, pg*
*926, 2022)*. Nature, 2023.
- 23. Levy, J.M.M. and A. Thorburn, *Autophagy in cancer: moving from understanding mechanism to*
*improving therapy responses in patients*. Cell Death and Differentiation, 2020. **27**(3): p. 843-
857.
- 24. Jin, Y., et al., *Depletion of Adipocyte*
*Leads to Lipodystrophy and Metabolic Dysregulation*. Diabetes, 2021. **70**(1): p. 182-195.
- 25. Park, H.W., et al., *Alternative Wnt Signaling Activates YAP/TAZ*. Cell, 2015. **162**(4): p. 780-794.
- 26. Liu, T.Y., et al., *The β -catenin/YAP signaling axis is a key regulator of melanoma-associated*
*fibroblasts*. Signal Transduction and Targeted Therapy, 2019. **4**.
- 27. Yuan, Y., et al., *YAP1/TAZ-TEAD transcriptional networks maintain skin homeostasis by*
*regulating cell proliferation and limiting KLF4 activity*. Nature Communications, 2020. **11**(1).
- 28. Jiang, L.Y., et al., *YAP-mediated crosstalk between the Wnt and Hippo signaling pathways*.
Molecular Medicine Reports, 2020. **22**(5): p. 4101-4106.

- 29. Li, N.S., N.H. Lu, and C. Xie, *The Hippo and Wnt signalling pathways: crosstalk during*
*neoplastic progression in gastrointestinal tissue*. *Febs Journal*, 2019. **286**(19): p. 3745-3756.
- 30. Deng, F.H., et al., *YAP triggers the Wnt/ β -catenin signalling pathway and promotes enterocyte*
*self-renewal, regeneration and tumorigenesis after DSS-induced injury*. *Cell Death & Disease*,
2018. **9**.
- 31. Zi, Y.Y., et al., *Phosphorylation of PPDPF via IL6-JAK2 activates the Wnt/ β -catenin pathway in*
*colorectal cancer*. *Embo Reports*, 2023. **24**(9).
- 32. Guo, X.L., et al., *IL-6 accelerates renal fibrosis after acute kidney injury via DNMT1-dependent*
*FOXO3a methylation and activation of Wnt/ β -catenin pathway*. *International*
*Immunopharmacology*, 2022. **109**.
- 33. Shi, H., et al., *Lipocalin 2 promotes lung metastasis of murine breast cancer cells*. *Journal of*
*Experimental & Clinical Cancer Research*, 2008. **27**.
- 34. Yang, J., et al., *Lipocalin 2 promotes breast cancer progression*. *Proceedings of the National*
*Academy of Sciences of the United States of America*, 2009. **106**(10): p. 3913-3918.
- 35. Chaudhary, N., et al., *Lipocalin 2 expression promotes tumor progression and therapy*
*resistance by inhibiting ferroptosis in colorectal cancer*. *Int J Cancer*, 2021. **149**(7): p. 1495-
1511.
- 36. Lauby-Secretan, B., et al., *Body Fatness and Cancer--Viewpoint of the IARC Working Group*.
*N Engl J Med*, 2016. **375**(8): p. 794-8.
- 37. Recalde, M., et al., *Longitudinal body mass index and cancer risk: a cohort study of 2.6 million*
*Catalan adults*. *Nature Communications*, 2023. **14**(1).
- 38. Rybinska, I., et al., *Cancer-Associated Adipocytes in Breast Cancer: Causes and*
*Consequences*. *International Journal of Molecular Sciences*, 2021. **22**(7).
- 39. Zhu, Q.Z., et al., *Adipocyte mesenchymal transition contributes to mammary tumor progression*.
*Cell Reports*, 2022. **40**(11).

REVIEWERS' COMMENTS

Reviewer #1 (Remarks to the Author):

The revised study has been significantly improved. The reviewer appreciates the efforts of the authors to address the questions raised before.

One suggestion for the newly included luciferase reporter data (Fig. 6j-k):

- The predicted TEAD binding sites in L3 should be further mutated to strengthen the conclusion.

Reviewer #2 (Remarks to the Author):

The authors have carefully addressed all my questions by both new experiments and new discussions. I do not have more concerns, and suggest to consider the manuscript for publication in Nature Communications.

Kai Sun

Reviewer #3 (Remarks to the Author):

I recommend this paper for publication.

The authors have made a considerable effort in order to address the concerns of all 3 reviewers. The addition of further in vivo data using the colon cancer model is helpful, as are some of the mechanistic experiments i.e. ChIPseq investigating TEAD box binding, and investigating autophagy versus BECLIN specific functions.

Note: figures need to be edited to comply with the journal's requirements to show scatterplots (rather than bar graphs) for data where the $n < 10$, including in the new data added. Additionally some of these new graphs need bars with *s (or ns) to indicate significance easily as well.

Fig 3b-e should have an insert panel at higher magnification because it is challenging to determine colocalization at the current magnification.

REVIEWERS' COMMENTS

Reviewer #1 (Remarks to the Author):

The revised study has been significantly improved. The reviewer appreciates the efforts of the authors to address the questions raised before.

We express great appreciation to the reviewer for the positive evaluation.

One suggestion for the newly included luciferase reporter data (Fig. 6j-k):

- The predicted TEAD binding sites in L3 should be further mutated to strengthen the conclusion.

We thank the reviewer for the additional suggestion to strengthen our work. In response, we deleted the TEAD binding sites from our original luciferase constructs and detected luciferase activity (Revised Fig. 1a). The result revealed a marked reduction in LCN2 promoter activity in the absence of TEAD binding motifs, indicating that TEAD-YAP directly amplifies LCN2 expression (Revised Fig. 1b).

Revised Fig. 1

The main Figures 6j-k have been replaced

Reviewer #2 (Remarks to the Author):

The authors have carefully addressed all my questions by both new experiments and new discussions. I do not have more concerns, and suggest to consider the manuscript for publication in Nature Communications.

Kai Sun

We are grateful for the opportunity to have our manuscript evaluated by the reviewer.

Reviewer #3 (Remarks to the Author):

I recommend this paper for publication.

We are grateful to have our manuscript evaluated by the reviewer.

The authors have made a considerable effort in order to address the concerns of all 3 reviewers. The addition of further in vivo data using the colon cancer model is helpful, as are some of the mechanistic experiments i.e. ChipSeq investigating TEAD box binding, and investigating autophagy versus BECLIN specific functions.

Note: figures need to be edited to comply with the journal's requirements to show scatterplots (rather than bar graphs) for data where the $n < 10$, including in the new data added. Additionally some of these new graphs need bars with *s (or ns) to indicate significance easily as well. Fig 3b-e should have an insert panel at higher magnification because it is challenging to determine colocalization at the current magnification.

We sincerely appreciate the editor's careful consideration to our work.

We have revised the figures to align with the journal's requirements.